# SoK: Evaluations in Industrial Intrusion Detection Research

## Abstract

Industrial systems are increasingly threatened by cyberattacks with potentially disastrous consequences. To counter such attacks, industrial intrusion detection systems strive to timely uncover even the most sophisticated breaches. Due to its criticality for society, this fast-growing field attracts researchers from diverse backgrounds, resulting in 130 new detection approaches in 2021 alone. This huge momentum facilitates the exploration of diverse promising paths but likewise risks fragmenting the research landscape and burying promising progress. Consequently, it needs sound and comprehensible evaluations to mitigate this risk and catalyze efforts into sustainable scientific progress with real-world applicability. In this paper, we therefore systematically analyze the evaluation methodologies of this field to understand the current state of industrial intrusion detection research. Our analysis of 609 publications shows that the rapid growth of this research field has positive and negative consequences. While we observe an increased use of public datasets, publications still only evaluate 1.3 datasets on average, and frequently used benchmarking metrics are ambiguous. At the same time, the adoption of newly developed benchmarking metrics sees little advancement. Finally, our systematic analysis enables us to provide actionable recommendations for all actors involved and thus bring the entire research field forward.

## 1 Introduction

The digitalization of Industrial Control Systems (ICSs) has led to an escalating rise in cyberattacks [5, 52, 67], of which prominent ones include the Stuxnet or Ukrainian power grid attacks. These attacks are boosted by widely deployed legacy devices not meant to implement crucial security measures [15]. Specialized Industrial Intrusion Detection Systems (IIDSs) address this gap by providing an easily retrofittable security solution for legacy industrial deployments [16, 27]. To this end, IIDSs passively monitor network traffic or the physical process state and alert human operators to initiate adequate countermeasures in case of suspected attacks [74].

As an emerging hot research area, IIDSs attract researchers and industrial operators from diverse backgrounds. It thus comes as no surprise that, according to our literature research, at least 1109 distinct authors have published ideas for detection mechanisms between 2019 and 2021 alone. While their diverse background is beneficial to cover lots of different perspectives and ideas, the resulting fast-paced advancements lead to a lack of established evaluation methodologies and comparability across the field. Consequently, worthwhile ideas remain hard to identify, and it is unclear which improvements are suitable to close the gap to much-needed production-ready IIDSs. Ideally, the vast research efforts would be channeled through clear, comparable, coherent, and expressive evaluation methodologies. Only through a resulting comparability between approaches can the IIDS research landscape fully benefit from its high diversity.

Digging deeper into conducted evaluations, researchers use benchmarking datasets that are either publicly available or, more commonly, custom-made for that specific test hindering repeatable experiments. Based on these datasets and an IIDS' alerts, various (performance) metrics are computed. However, IIDSs are often evaluated on pre-selected datasets, covering specific favorable scenarios [14]. Furthermore, metrics are chosen or designed based on specific goals determined (to some degree arbitrarily) by the researchers. The resulting custom evaluation methodologies lead to an immense heterogeneity within the IIDSs research landscape, where most works, despite common goals, lack comparability. Consequently, technological and scientific progress is inhibited.

In this regard, meta-analyses of IIDS research already unveiled inefficiencies in the detection capabilities of published works [17] or criticized the conclusions drawn from scientific evaluation procedures [8, 23, 43]. Simultaneously, we observe attempts to fix these issues by, e.g., collecting representative benchmarking datasets [14], inventing specialized industrial metrics to accurately assess the "success" of an IIDS [24, 32–35, 38, 40, 44, 69], or by providing an abstract format to facilitate a coherent research landscape [74]. However, related work so far still fails to (i) quantify how IIDSs are evaluated within the vast body of literature, (ii) assess the applicability and impact of recent critiques partially known from, e.g., traditional intrusion detection [8, 51, 66], and (iii) deliver overarching recommendations to pave the way towards the shared goal of improving IIDSs to truly protect industrial networks and critical infrastructure against future cyberattacks.

With this paper, we strive to close the outlined gap with a Systematization of Knowledge (SoK) on the evaluation methodologies across IIDS research. To this end, we conduct a Systematic Mapping Study (SMS) to quantify the current state of the research landscape encompassing 609 papers.

From the resulting knowledge basis, we can draw a clear picture w.r.t. positive and negative developments as well as persistent flaws. Ultimately, our works allow us to provide clear recommendations for all involved actors to catalyze their joint efforts to protect the world's most critical networks.

**Contributions.** To pave the way toward a more coherent IIDS landscape, we make the following contributions:

- We survey 609 papers published until 2021 proposing IIDS designs and extract information about how their respective evaluations were conducted (Sec. 3).

- We systematize the gained knowledge w.r.t. utilized datasets and metrics to identify positive and negative trends as well as their potential for future improvements. We then complement these *theoretical* results with *practical* experiments to extend the understanding of the interplay between datasets and metrics (Sec. 4 and Sec. 5).

- Finally, we summarize current flaws in IIDS evaluations and formulate recommendations to improve future IIDS research for all involved actors: IIDS researchers, dataset creators, and industrial operators (Sec. 6).

**Artifact Availability.** We make the data of our SMS publicly available at https://www.dropbox.com/sh/bvhlrinhv4rn50u/AAAmQxzzGqZmU-7E0yfRvxZXa, and will publish our evaluation tools used for the practical experiments upon acceptance (for anonymity purposes).

## 2 Research on Industrial Intrusion Detection

To lay the foundation for our work, we provide a brief introduction to the field of industrial intrusion detection (Sec. 2.1) and its challenges (Sec. 2.2) before we discuss related work on the evaluation methodologies of this research field (Sec. 2.3). Based on this, we motivate the need for systematizing the knowledge on evaluating industrial intrusion detection research and formulate basic research questions (Sec. 2.4) to ultimately steer future research in an effective direction.

### 2.1 Industrial Intrusion Detection

The high degree of digitization in industries unleashes an enormous level of automation by integrating sensors, actuators, and control logic into tightly coupled cyber-physical systems. The current trend to build ICSs by adapting once proprietary and local network protocols, e.g., Modbus, to ubiquitous Ethernet networks, e.g., using ModbusTCP, paired with connectivity to the Internet, enables unique applications, e.g., remote monitoring or Supervisory Control and Data Acquisition (SCADA). Yet, these technologies simultaneously open new attack vectors, as prominent attacks demonstrate [5, 52].

To counter these security issues, various preventive measures have been proposed, e.g., secure variants of industrial communication protocols [14, 15]. But, retrospectively integrating these measures into existing ICSs, operating for decades, is costly, if possible at all, due to their strict requirements toward, e.g., availability and latency. In this context, intrusion detection is proposed as a promising alternative or complementing technology to passively retrofit security into ICSs [74] by monitoring systems or networks for suspicious activities or violations of security policies. However, established intrusion detection solutions from computer networks serving, e.g., offices or data centers, are not as effective in industries [76], primarily due to ICSs' reliance on unique (real-time) hardware such as Programmable Logic Controllers (PLCs) and sophisticated, custom-tailored attacks targeting the physical process [5, 70]. Consequently, research focuses on specialized Industrial Intrusion Detection Systems (IIDSs), which leverage the repetitive and predictable characteristics occurring in, e.g., Modbus' communication patterns or the physical process.

The IIDS research landscape can be coarsely classified along five dimensions: attacker model, detection technique, benchmarking environment, evaluation metric, and reactions. The *attacker model* influences which kind of attacks an intrusion detection system should be able to detect and potentially even differentiate. Note that while some surveys consider fault detection similar to attacks [49], faults do not occur as a consequence of cyberattackss but rather through, e.g., wear and tear [27] and are thus left out of the scope of this work. Thus, the *attacker model* determines an IIDS's input data, with common ones being network traffic, host data from SCADA systems or PLCs, and physical process data [49].

The main work of researchers then goes into designing the actual *detection technique*, which can be loosely categorized into knowledge-based, behavior-based, or hybrid approaches [53]. While knowledge-based systems (also referred to as misuse or supervised detection [53]) identify harmful behavior based on (known) patterns, behavior-based IIDSs rather specify how the ICS behaves normally and alert deviations from usual actions. Moreover, the detection technique is also heavily influenced by the attacker model. While attacks on a network layer are best detected on a per-packet basis, e.g., with deep-packet inspection [29], process-based detection can leverage a broader view of the ICS, e.g., by analyzing whether the physical process moves towards a critical state [13].

To validate the design of a detection technique and facilitate comparability of a newly proposed IIDS, its detection performance is evaluated with the help of suitable *benchmarking environments* and *evaluation metrics* (potentially in addition to computational performance or w.r.t. explainability [64]). Despite the data type, benchmarking environments for all kinds of industrial domains come in different forms, such as datasets, physical testbeds, simulations, or real facilities [14, 36]. Each type has its own trade-offs in terms of, e.g., accessibility, cost, or closeness to real deployments, so their selection needs to be carefully made. Moreover, the IIDS'

performance needs to be measured based on sensitive metrics. In that regard, scientists can refer to a plethora of common metrics [61] expressing the amount of false positive alerts or more complex characteristics (cf. Appx. B).

A final dimension is the *reaction* to IIDS alerts to mitigate an attack. Especially when transferring an IIDS to real-world deployments, operators may conduct (manual) forensic analyses to understand the cause for alert [4] and ultimately mitigate the threat [67] by, e.g., applying firewall rules. Preventive measures can also be coupled directly to a detection mechanism for more automated reactions, then called intrusion prevention systems. Those do, however, need to be carefully designed, since in an industrial setting simply blocking suspicious traffic may cause more harm than the attack itself.

## 2.2    Challenges of Evaluating IIDS

IIDS research takes place in a diverse field encompassing ICS architectures ranging from water supply over power delivery to manufacturing, where cyberattacks are primarily unique to a particular deployment [5, 52]. Even though ICSs rely on researchers to design appropriate countermeasures and test their efficiency in real-world deployments, operators rarely provide such urgently-needed data samples [3, 50, 66]. While these challenges constitute an opportunity to tackle IIDS research from varying angles, transfer insights across industrial domains, and investigate their efficiency in real-world deployments, they likewise segregate the overall research landscape, resulting in isolated silos [74]. Consequently, sound scientific evaluations remain as the foundation to facilitate coherence and measure the overall progress of the research field.

However, due to influences from various fields and a generally high interest in IIDSs, so far no coherent evaluation methodology could be established and subsequently improved. In practice, the path taken by most researchers to design and test their IIDSs relies on privately acquired and/or public (synthetic) datasets containing samples of benign traffic and/or physical process data as well as attack scenarios. To evaluate their IIDSs, researchers first train (and configure) their IIDS on samples of *benign* behavior and/or *attacks* (depending on the type of IIDS) from a specific industrial scenario. On a second evaluation dataset, they then compare the IIDS output (alerts) to the attack labels contained within the chosen dataset, i.e., they track how well the IIDS detects attacks and to which degree benign traffic or process values are unintentionally classified as suspicious. Finally, various metrics, e.g., the F1 score, quantify the detection performance and serve as the basis for comparisons to related work.

While most works adhere to this loosely outlined evaluation methodology, the devil is in the details [43]. Optimally, a given dataset would be suitable for a large amount of IIDS types and thus constitute a reference benchmark. However, widely-used datasets usually cover only specific industrial domains and a small subset of imaginable attacks [14]. Thus, the

datasets made available to the research community decisively influence the scenarios within which IIDSs are evaluated and also the types of attacks IIDSs are optimized for. Moreover, utilized evaluation metrics do not draw a complete picture of an IIDS's detection performance without putting them into context [27], which rarely happens adequately within the research field. As a matter of fact, this lack of hardened and proven research methodologies has been exposed to various criticism in recent years, as identified by related work.

## 2.3    Related Work on Evaluating IIDSs

Taking a closer look at recent literature on the challenges of evaluating industrial intrusion detection research (cf. Sec. 2.2), we identify a range of works discussing and criticizing the current state of IIDS research. First, various surveys provide an overview of the utilized *detection methods* across that research field [16, 27, 49, 53, 63, 67, 74, 75], ranging from learning specific communication patterns to analyzing the physical state of the monitored system. In this context, difficulties reproducing results and generalizing IIDSs to related ICSs domains beyond those specifically evaluated were reported [17, 74]. While these surveys repeatedly cover more than 70 publications, showing the huge attention industrial intrusion detection attracts, at the same time, they indicate a lack of coherence and advancement within the research field.

Similar surveys focused on summarizing available *datasets* and testbeds (from which datasets can be generated) specifically designed for IIDS evaluations [14, 36]. These efforts identify at least 61 testbeds and 23 benchmarking datasets that are publicly available [14]. Since these surveys focus solely on datasets, they lack essential analyses about the actual application of datasets. As a rare exception, Balla et al. [10] analyzed dataset usage for deep learning detection methodologies, observing a strong bias toward non-ICS datasets, such as the KDD dataset family, with a usage of over 50 %.

Besides the used dataset or testbed, the choice of *metrics* plays an important role when evaluating IIDSs. Without a dedicated focus on *industrial* intrusion detection, Powers [61] provides an overview of different metrics and puts their expressiveness into context. Yet, the considered point-based metrics (cf. Appx. B.1), e.g., accuracy or precision (also used in other domains such as machine learning), must be used carefully not to introduce any biases [61]. Moreover, especially for evaluations on (industrial) time-series datasets, further challenges, such as an imbalanced representation of attacks, have to be considered [8, 25]. Consequently, more advanced time series-aware metrics have been proposed [24, 32–35, 38, 44, 69] (cf. Appx. B.2). While this development promises to enhance the expressiveness of evaluations, their soundness and usage remain mostly unexplored so far.

Finally, various *meta-surveys* focus on machine learning pitfalls for industrial intrusion detection [18, 23, 50, 63] or highlight challenges when transferring IIDSs from research

to actual industrial deployments [3, 50, 66]. These problems include, e.g., inappropriate use of metrics [8], the dominance of lab-based datasets [8, 63], or predominant focus on only a few of the wide range of industrial domains and protocols [63]. Importantly, empirical data on the evaluation of IIDS research is not yet available.

In summary, evaluations of IIDSs can, in theory, be based on a solid foundation of public datasets and advanced metrics. However, this research branch lacks a decent understanding of the methodologies actually applied within it beyond individual criticism regarding isolated aspects.

## 2.4 The Need for Systematization

The tremendous research interest in industrial intrusion detection, with 130 publications in the year 2021 alone, has led to a huge variety of evaluation methodologies. The resulting fast-paced research has a huge risk of becoming disjoint [74], eventually slowing down the overall progress in securing ICSs. Most importantly, the heterogeneity across industrial domains [74] and an observed widespread evaluation bias [27, 70, 74] make comparisons between IIDSs difficult. Past surveys on detection methodologies, datasets, metrics, and meta-studies have only studied individual aspects in isolation from each other (cf. Sec. 2.3). Thus, to unveil the root causes hindering coherent and sustainable IIDS research, there is a need to systematically consolidate the current state of evaluations in industrial intrusion detection research to ultimately identify remedies against the status quo.

We argue that only by analyzing how IIDSs are evaluated on a broad scale, as done in a Systematic Mapping Study (SMS) [41], we can comprehensively tackle the question of research coherence and evaluation soundness, i.e., to which extent evaluations are performed on uniform (public) datasets with widespread and suitable metrics to achieve a high level of comparability. More precisely, we aim to answer the following research questions:

▶ **Q1:** Which datasets are actually used to evaluate IIDSs?
▶ **Q2:** To which extent do IIDSs compare against each other?
▶ **Q3:** Which metrics are utilized in evaluations of IIDSs?

Besides providing a comprehensive picture of the traits and characteristics of IIDS evaluations, answering these questions lays the foundation to formulate actionable recommendations for IIDS evaluation, enabling the different actors within the research community to focus their joint effort on the overarching challenge of securing industrial deployments.

## 3 Systematic Mapping Study

The objective of this SoK is to provide a systematic understanding of how (differently) IIDS research is currently evaluated and how this current status quo can be sustainably improved. While related work already hints at prevalent issues that might prevent objective comparisons (cf. Sec. 2.3), a

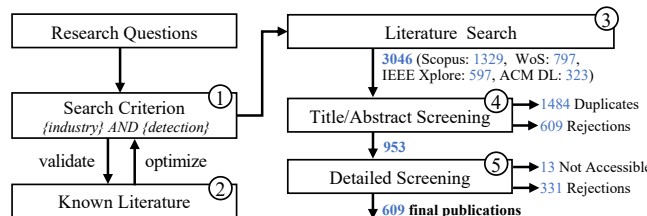

Figure 1: To conduct the SMS, we follow a two-staged approach which results in extracting a total of 609 relevant publications proposing novel IIDS as of December 2022. We list the corresponding search string in Appx. A.

holistic analysis is missing so far. Therefore, we strive to ascertain the state of IIDS evaluation methodologies by conducting a Systematic Mapping Study (SMS), a variation of a classical Systematic Literature Review (SLR) [41], to obtain a large, qualitative, and unbiased collection of relevant publications in a verifiable process, oriented along established best practices and guidelines [41]. First, we search relevant papers for a broad subject (IIDSs proposals) from the scientific literature with a systematic process. Afterward, publications are analyzed and classified based on the subjects of our analysis (Q1–Q3), i.e., their evaluation methodology.

To holistically answer the outlined research questions for a large and heterogeneous research field, we perform a comprehensive SMS as depicted in Fig. 1. According to the research questions, the SMS focuses on publications that propose IIDSs for ICSs as researchers naturally have to evaluate their performance in a scientific manner. In contrast to Balla et al. [10], we only consider publications that leverage at least one *industrial*-specific dataset, i.e., they were obtained from an ICS, e.g., include specific protocols such as Modbus, physical process data, or ICS-specific cyberattacks.

To conduct our SMS, we leverage Parsifal [21] to organize and comprehensibly document our screening process. First, we transformed the research questions into a search string ① (cf. Appx. A), which we successively optimized through validation with an initial set of known and representative literature ②. We then queried four search engines (IEEE Xplore, ACM DL, Scopus, and Web of Science) on December 2022 and found a total of 3046 hits ③. From this initial set of publications, we discarded duplicates (1484 publications) and performed a first screening of all remaining publications' titles and abstracts ④. In this initial screening, we mostly focused on removing publications from other research domains that still matched our search string and such publications that clearly do not propose (and thus evaluate) an IIDS approach. After this first screening phase, 953 unique publications remained for further consideration. Note that we did not filter for any specific detection techniques. Still, most publications covered by the survey (and thus the research field) resemble machine-learning.

In a final step, we conducted a detailed screening of the re-

maining publications to extract those that build the foundation for our further analysis ⑤. When accessing the full text of all papers, only 13 publications were not accessible to us and thus omitted. We performed a detailed second screening of all remaining publications, resulting in 331 further rejections of those that do not match our requirements for proposing IIDSs, e.g., belonged to fault detection (cf. Sec. 2.1). From the resulting set of 609 accepted publications, we extracted the relevant data to answer our research questions, such as the datasets and metrics they utilize for their evaluations. To ensure consistency, one author performed the detailed screening and data extraction while the workload for initial title/abstract screening was shared across multiple persons.

Through our systematic approach, to the best of our knowledge, we are the first to analyze the entire IIDS landscape. With 609 analyzed publications, our work is based on a significantly larger knowledge base than any of the previous surveys of related work (cf. Sec. 2.3). This basis enables us to analyze the evaluation methodologies of the broad IIDS research landscape. Beyond presenting our findings, releasing our SMS as a public artifact (cf. Artifact Availability) may help future researchers to find appropriate candidates for comparisons, facilitates further analyses, or enables tracking of the progress within the ICS domain in the future.

## 4 IIDS Evaluation in Research

With a systematic basis of 609 publications proposing IIDSs gathered in our SMS (cf. Sec. 3), we now assess how the overall research landscape on evaluation methodologies for IIDSs has evolved over time. As a systematic representation has been missing so far (cf. Sec. 2.3), we augment the field with a high-level overview in Sec. 4.1. Afterward, we unveil common trends in evaluation methodologies, especially w.r.t. the utilized datasets (cf. Sec. 4.2). Finally, we study the degree of comparability between IIDSs publications in terms of the utilized dataset and evaluation metric (cf. Sec. 4.3).

### 4.1 Overview of the IIDS Research Landscape

We begin our analysis with a high-level overview of the evolution and composition of the IIDS research landscape.

#### 4.1.1 Evolution

To understand the evolution of the IIDS research domain, we focus on the number of published papers over time (cf. Fig. 2), which we enrich with timestamps of notable cyber incidents and the releases of commonly used evaluation datasets. While the first publications within the IIDS domain date back to 2003, the domain initially received little attention, with only 28 publications until 2012. From 2013 onward, research took off exponentially, with an average increase of 40.9 % in yearly publications. In 2021, the last year considered in our SMS,

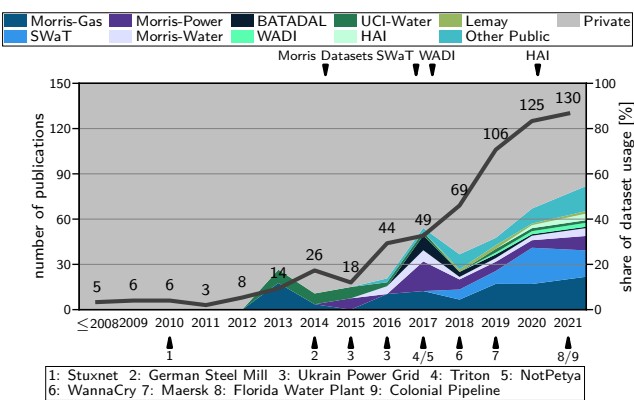

Figure 2: Publications on IIDSs took off around 2013 and kept increasing as more cyberattacks occurred. Simultaneously the trend fosters to evaluate on public datasets.

we identified 130 new publications, which is higher than in any previous year. In comparison, the Top 10 cyber security conferences experienced a lower average yearly increase in accepted publications from 7.2 % for Crypto up to only 25.5 % for USENIX Sec during the same timespan [77].

We presume that the key driver for this development and interest in this research domain is caused by the raised public awareness following the Stuxnet cyberattack and subsequent ones like the two major incidents with the Ukrainian power grid [5]. Apart from such targeted attacks, industries were equally affected by more widespread malware, such as Not-Petya or WannaCry [5], due to their increasing digitalization and Internet-facing deployments (cf. Sec. 2.1). With attacks still continuing [52], endangering human safety, expensive equipment, as well as the environment, the peak in 2021 with 130 proposals comes as no surprise—underlining the growing importance of IIDS research.

A first look at the (publicly) utilized datasets' in Fig. 2 also allows us to deduce the existence of a growing number of public datasets. These datasets stem from various industrial domains, such as water purification, gas distribution, and electrical power generation, among many others. This conclusion aligns with recent results identifying a growing number of public datasets emerging across many industrial domains [14].

From this initial assessment, we conclude that IIDS research tackles the diverseness of industrial domains based on variously utilized datasets and experiences steady growth that does not seem to have reached its peak yet.

#### 4.1.2 Coherence

For such a rapidly growing research landscape in a diverse industrial environment, we further want to understand how coherent research is performed, i.e., whether directions exist that receive more attention and whether recent results build on previous findings. Therefore, we visualize the connections

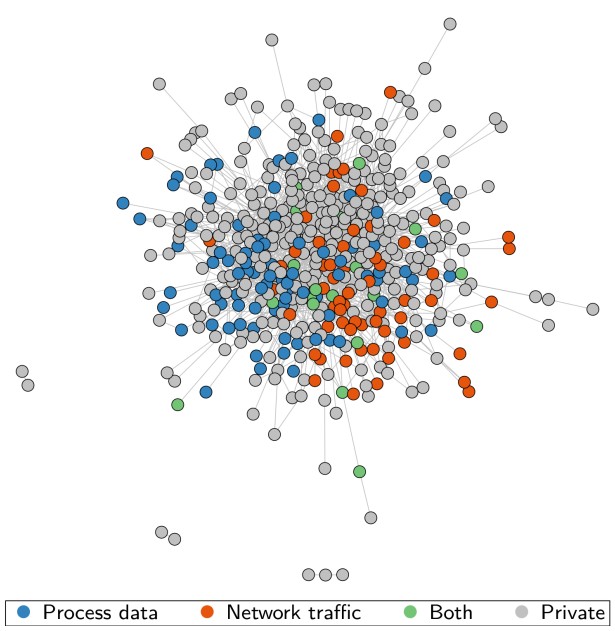

Process data ● Network traffic ● Both ● Private

Figure 3: Publications arranged in a citation graph reveal two directions roughly disjunct into approaches considering network traffic datasets and ones evaluating process data.

| Origin | Name | Type | Domain | Protocol | Usage |
|---|---|---|---|---|---|
| iTRUST[a] | SWaT [28] | P* | Water | – | 9.0 % |
| | BATADAL [68] | P | Water | – | 1.6 % |
| | WADI [2] | P | Water | – | 1.0 % |
| Morris et al.[b] | Morris-Gas [55] | N | Gas | Modbus | 11.8 % |
| | Morris-Power [1] | P | Electricity | – | 5.6 % |
| | Morris-Water [55] | N | Water | Modbus | 2.8 % |
| Misc | UCI-Water [60] | P | Water | – | 2.0 % |
| | HAI [65] | P | Diverse | – | 1.1 % |
| | Lemay [45] | N | Electricity | Modbus | 1.0 % |

N: Network captures       P: Process data

* Network captures for SWaT exist, but are rarely used in research.
[a] https://itrust.sutd.edu.sg/itrust-labs_datasets
[b] https://sites.google.com/a/uah.edu/tommy-morris-uah/ics-data-sets

Table 1: Across the top nine public datasets, two account for the majority of uses. Despite ICSs' diversity, the top datasets focus on a few domains and protocol combinations.

among publications by their citation relationships in Fig. 3. Citation data was retrieved and aggregated from OpenAlex and Semantic Scholar for all 609 publications, and we draw a connection between two publications if one cites another. In Fig. 3, publications are arranged by the force-directed Fruchterman-Reingold placement algorithm [22], i.e., connected vertices are pulled closer together. Moreover, for publications utilizing publicly accessible datasets, we colored their vertices belonging to process data datasets, network traffic, or both. Note, however, that our analysis omits 125 publications for which no connection to other publications could be found, either because the citation data for the respective publications was incomplete or because the IIDSs were indeed presented without relating to the vast body of existing works.

On average, a publication is cited by 2.9 other IIDS publications, while the Top 5 cited publications [13, 29, 30, 42, 70] (not in order) are cited on average by 46.6 papers as of the 1st March 2023. These numbers provide a first glance at the connectivity in IIDS research.

Yet upon an initial inspection of the citation structure, we observe that the IIDS research domain is divided into two basic directions based on the evaluated dataset types: A first group of 102 papers (blue) resembles the larger class that focuses on process data datasets. In addition, we discovered a slightly smaller class of 81 publications (red) that corresponds to intrusion detection methodologies detecting attacks in network data. Only rarely (19 times) do IIDSs fall into both classes (green). Interestingly, both research fields show

little connectivity, indicating a limited exchange of knowledge across these fields. This is backed by the fact that the clustering coefficient for the sub-domains (process data 0.15 and network traffic 0.13) is slightly higher than for the entire IIDS research landscape (0.11).

Consequently, publications are more likely to cite each other if they stem from the same type, which promises a high number of comparisons among them. Still, the low clustering indicates incoherence in the overall research domain.

## 4.2 Benchmarking Datasets

With a basic understanding of the IIDS research domain, we now assess how evaluations are conducted in more detail. In this context, the chosen benchmarking datasets are a crucial building block as it serves as the basis for nearly all subsequent performance calculations. While related work has assessed which datasets are readily available [14], their exact usage and distribution remains unknown as of now (cf. Sec. 2.3). Consequently, this section answers our first research question Q1, regarding the datasets IIDSs are evaluated on. For a description of the existing datasets and testbeds, please refer to the survey conducted by Conti et al. [14].

### 4.2.1 Overview

As can be derived from Fig. 2, over the entire timespan, the majority of used datasets are private, and only 33.3 % of the publications evaluate at least one public dataset. Note that we counted datasets as private if there existed no obvious procedure to retrieve the dataset. While private datasets may represent unique use cases, e.g., real-world data of industrial facilities, they significantly hinder reproducibility and comparisons to related works since they usually deny access to

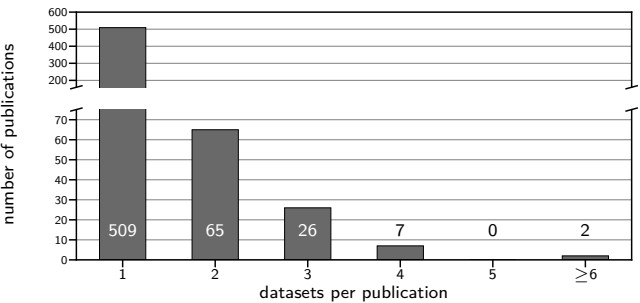

Figure 4: Publications usually utilize a single dataset, and only 16.4 % of the papers leverage multiple datasets at all.

| Combination | Count | Origins |
|---|---|---|
| Morris-Gas & Morris-Water | 12 | 1 |
| Morris-Gas & Morris-Power | 8 | 1 |
| Morris-Power & Morris-Water | 7 | 1 |
| SWaT & WADI | 4 | 1 |
| Morris-Gas & UCI-Water | 4 | 2 |
| Morris-Gas & SWaT | 3 | 2 |
| Electra Modbus & S7Comm | 3 | 1 |
| Morris-Gas, Power & Water | 5 | 1 |

No private datasets were considered

Table 2: If multiple datasets are used, they mostly stem from the same class or origin, attributing little to richer evaluations.

outsiders. In our SMS, we refrained from investigating private datasets in more depth because of the varying degrees of descriptions throughout the publications. Hence, needed details cannot be fully captured or verified. Nonetheless, we observe a trend starting around 2013 toward increased utilization of public datasets, which accounts for 54.7 % of the evaluated datasets in 2021. Therefore, it is more likely that an IIDS uses public datasets if published recently.

This trend follows the publication of high-quality datasets that are still widely used today. When looking at peak usage of public datasets, the SWaT [28] and Morris-Gas Pipeline [55] datasets jointly occur in 20.4 % of the publications, which is the majority of the publications utilizing a public dataset at all (33.3 %) and other public datasets are thus used much less frequently. As a consequence, a significant portion of research activities seems to be biased toward these two datasets.

Regarding dataset diversity, across our entire SMS, we identified 35 unique public datasets, which exceeds previous reports of 23 datasets by Conti et al. [14]. In contrast to Balla et al. [10] (cf. Sec. 2.3) and by the design of our SMS (cf. Sec. 3), we dominantly encounter specialized industrial datasets contradicting their observed research bias toward non-industrial datasets. However, of the many public datasets, 16 are only used once, and 14 occur at least three times (the Top nine public datasets are depicted in Fig. 2). Thus, availability alone is not decisive for a widespread use and other factors such as covered domain and attacks as well as the overall quality of the data seems to play an essential role as well.

### 4.2.2 Dataset Types

In the next step, we examine the Top nine datasets more closely and highlight their different directions (cf. Tab. 1).

First, a dataset's *type* can be either a network capture, mostly required for network-based IIDSs or a (preprocessed) sample of physical system data, e.g., a time series of temperature values. For each type, we observe one major origin that accounts for most of the utilization across research, with iTRUST for process-based datasets and Morris et al. primarily for network-based ones. Considering the type of the top nine

utilized datasets, we observe a strong focus on process-based datasets with 20.3 % compared to 15.6 % for network-based, which is in line with the observations from Sec. 4.1.2.

Since industrial domains are diverse, we expect a large coverage of them across utilized datasets as well. However, the commonly covered industrial domains are mainly driven by the water and gas facilities, indicating an underrepresentation of all other domains, such as power generation, electricity distribution, or manufacturing. Yet, considering the large numbers of domains covered by private datasets, for which (high-quality) public alternatives do not exist, we cannot conclude that other domains receive few attention nor that those industries show no interest in IIDS research.

Lastly, industries are well known for their diverse and incompatible pooling of network protocols, mostly for legacy reasons [15]. Despite market-share studies identifying 11 dominant network technologies [31], research either focus on Modbus (having 10 % market share [31]) or no communication protocol at all. While we discovered IIDSs for further industrial protocols such as IEC 60870-5-104 [46], S7 [47], or DNP3 [62], their representation is marginal and mostly confined to private datasets. Therefore, the distributions of utilized datasets w.r.t. their type, industrial domain, and network protocol reveal a significant drift between peer-reviewed literature and actual production systems.

### 4.2.3 Research Embedding

In the last step, we assess how the different datasets are embedded into research. Therefore we begin with the number of different datasets that are used within a single publication, as shown in Fig. 4. A large class of publications (509) evaluates a single dataset, and only a minority (100) on more than one. One publication uses 1.3 datasets on average. This observation is in line with the previous clustering observed in Sec. 4.1.2, which is more coherent w.r.t. the top-used datasets, suggesting that researchers often primarily focus on a single dataset. Given that we found at least 35 datasets publicly available, researchers most likely could consider additional, compatible datasets, especially when claiming that

proposed IIDSs are applicable to a large range of industrial domains [74]. This claim is backed by the fact that two publications have already evaluated as many as six datasets [11,26]. However, our results also suggest a discrepancy between datasets w.r.t. ease of use, documentation, and completeness, motivating the limited use of the available datasets.

Looking into the preferred datasets, Tab. 2 enumerates the top dataset combinations. While we observe prominent combinations, the corresponding datasets usually originate from the same source and thus represent similar domains and protocols. Only seven publications evaluate datasets that stem from two origins. Thus, potentially widely applicable IIDSs are evaluated for specific (research) deployments from a single industrial domain, most likely not representative of an entire domain. Consequently, research fails to effectively widen the scope of available evaluations and rather introduces biases by focusing on a few specific niches.

Overall, IIDS research is still governed by private datasets, with a steadily increasing trend toward public datasets. However, we observe the potential for improvement in the number of datasets used during evaluation as well as their diversity w.r.t. their type, industrial domain, and network protocol.

## 4.3 Reproducibility and Comparability

Next, we address our second research question Q2 asking to which extent IIDSs compare against each other. We assess this question from two directions, first by examining the conditions for reproducibility and second by measuring the degree of comparability, which are both perceived as good scientific standards [56], even though reproducibility lacks far behind expectations in the entire research community (beyond intrusion detection research) [9]. While reproducibility enables researchers to comprehend, build upon, or even enhance existing work, comparability allows them to determine how well an approach performs, i.e., to highlight the impact of newly proposed contributions over previous work or which approaches might be suitable for real-world deployments.

### 4.3.1 Reproducibility

Within IIDS research, reproducing existing work is not uncommon, e.g., to concisely analyze the prospects and limitations of individual approaches [17,43], prove the feasibility of new ideas upon reproduced implementations [74], or solely for scientific profoundness [56]. Yet, successfully reproducing approaches is not guaranteed [17]. To even enable the cumbersome process of reproducing IIDS research, the availability of artifacts, such as datasets or code, is needed.

In our survey, we observe that 33.3 % of the publications already utilize public datasets with an improving trend (54.7 % of utilized datasets in 2021 are public; cf. Fig. 2). However, successfully reproducing older publications is less likely. While the availability of code is not strictly required, as the

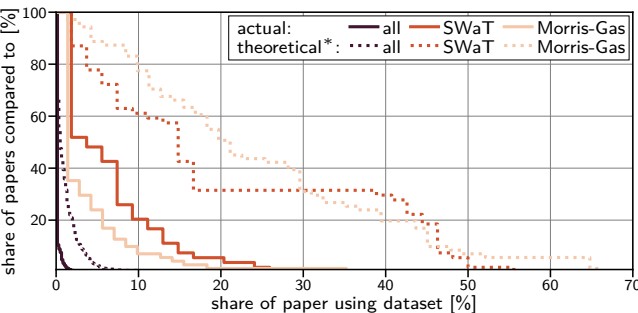

*A paper is comparable to all papers that share one dataset and metric and were published at least one year earlier.

Figure 5: On average, authors compare IIDSs to 0.5 approaches from the related work (black), while theoretically, they could compare to at least 6.0. This gap increases for papers evaluating the SWaT or Morris-Gas datasets.

relevant details should be part of the publication, it greatly eases the reproducibility process. Unfortunately, it is difficult to ascertain the availability of source code in a systematic way as it is not always clear where to find availability statements or corresponding pointers in publications. Still, we only encountered 21 publications with obvious references, e.g., clearly highlighted repositories. We subjectively deduce an overall low availability of source code across IIDS research.

Thus, researchers often have to rely solely on the descriptions and evaluation results provided by the paper to verify their code. Overall, reproducibility is thus challenging as optimally both criteria (public dataset and source code) have to be met. The increasing use of public datasets promises improvements in at least one direction, while publicly available artifacts accompanying publications remain the exception.

### 4.3.2 Comparability

Fortunately, cumbersome reproducibility is often not needed when, for example, it suffices to compare results to related work, e.g., to prove a novel attack detection approach superior. This requires that both works have been evaluated on at least one common dataset. Likewise, to objectively judge their detection performance, both publications must employ at minimum one identical evaluation metric. Common metrics include, but are not limited to, e.g., accuracy, precision, recall, or F1 [61]. Appx. B provides descriptions of further metrics.

To judge the degree of comparability across the research landscape for each publication, we extracted the *actual* number of comparisons made by the authors and calculated the number of *theoretically* possible comparisons. Therefore, while conducting our SMS (cf. Sec. 3), we gathered how many publications each author uses as comparison references and additionally extracted the exact metrics used in each publication's evaluation. We estimate the minimum amount of theoretically possible comparisons by counting a publication

as comparable if it shares at least one common dataset and metric and was published in an earlier year. Note that while not every two publications assume the same attack model, comparability can still be justifiable in the cases where the dataset matches since authors should select a dataset that best fits their approach. This methodology provides a great opportunity to assess actual and theoretical possible comparability, and Fig. 5 depicts the degree of comparability.

Overall, the number of *actual* comparisons performed by researchers is low, with 0.5 publications on average. For the two most-common datasets, we observe higher values (SWaT 2.4 and Morris-Gas 1.7). Still, there exists the *theoretical* opportunity for authors to compare a proposed IIDS to an average of 6.0 alternatives. On the one hand, this proves that many works are indeed comparable in terms of datasets and metrics. On the other hand, prominent datasets help in that regard since their theoretical comparability is higher (SWaT 10.0 and Morris-Gas 16.4). Note that it should not be the ultimate goal to compare against as many publications as possible since quality is preferential before quantity.

Looking closer into the details of Fig. 5, it is interesting that 10 % of the publications evaluating the Morris-Gas dataset (yellow) actually compare only against 7 % of different Morris-Gas publications. However, for SWaT (red), 10 % of publications are actually compared to about 18 % of existing works. Meanwhile, theoretical comparability for Morris-Gas publications is even higher than for SWaT (dotted lines). Regarding all publications (black), a total of 95.4 % of publications are not compared to a single IIDS.

The state of comparability in the IIDS research is decent but with opportunities for improvement in the future as many publications share common datasets and metrics already.

# 5 Survey on Evaluation Metrics

Previously, we analyzed comparability as a combination of utilizing overlapping datasets and evaluation metrics and observed that more publications could compare against each other in theory (cf. Sec. 4.3.2). However, our analysis still lacks a more detailed look at evaluation metrics used in IIDS research. Moreover, and most importantly, it is still unclear how expressive a given (combination of) metric(s) is in judging the detection performance of an IIDS.

To this end, we provide an overview of common and newly proposed metrics and categorize them into a taxonomy (cf. Sec. 5.1). Next, we assess their utilization across IIDS research along our SMS (cf. Sec. 5.2). Finally, since there exist known flaws to metrics (cf. Sec. 2.3), we examine how susceptible the research domain is in that regard by analyzing their expressiveness in practical experiments (cf. Sec. 5.3).

| | Metric | Appendix ● | TP | TN | FP | FN | Synonym |
|---|---|---|---|---|---|---|---|
| Point-based | TPR | B.1.2 | ✓ | | | ✓ | Recall Sensitivity Hit-Rate |
| | FNR | B.1.3 | ✓ | | | ✓ | Miss-Rate |
| | TNR | B.1.4 | | ✓ | ✓ | | Specificity Slectivity |
| | FPR | B.1.5 | | ✓ | ✓ | | Fall-out |
| | PPV | B.1.6 | ✓ | | ✓ | | Precision Confidence |
| | NPV | B.1.7 | | ✓ | | ✓ | – |
| | Accuracy | B.1.8 | ✓ | ✓ | ✓ | ✓ | Rand Index |
| | F1 | B.1.9 | ✓ | | ✓ | ✓ | – |
| | RoC | B.1.10 | ✓ | ✓ | ✓ | ✓ | – |
| | AuC | B.1.11 | ✓ | ✓ | ✓ | ✓ | – |
| Time-aware | Detected Scenarios | B.2.1 | ✓ | | | | – |
| | Detection Delay | B.2.2 | ✓ | | ✓ | | – |
| | (e)TaPR [33, 34] | B.2.3 | ✓ | ✓ | ✓ | ✓ | eTaP eTaR eTaF1 |
| | Affiliation [32] | B.2.4 | ✓ | ✓ | ✓ | ✓ | – |

Table 3: Our taxonomy distinguishes between point-based and time series-aware metrics. Metrics may occur under different synonyms. For details, refer to Appx. B.

## 5.1 A Taxonomy of IIDS Evaluation Metrics

Evaluating the performance of an IIDS is of utmost importance to prove its effectiveness and compare it quantitatively against related works either in terms of attack detection performance, or computational resources.

Since *computational* resources are stated only occasionally throughout the SMS, we shorty introduce which aspects were evaluated. The most prominent aspect, still in 136 publications, refers to the time to train a model or classify a given datapoint/dataset. More infrequently are statistics about CPU/GPU usage (13), RAM utilization (12), or model size (16). However, a sound comparison without equivalent hardware or implementations is challenging and therefore those metrics are beyond the scope of the SoK in the following.

Regarding *detection* performance, during the conduction of our SMS, we extracted a total of 167 distinct metrics that were used during the evaluations. To provide an initial holistic overview, we present the most used metrics found in the SMS and relevant (newer) ones observed in related work in a taxonomy (cf. Tab. 3). The metrics are discussed in a more general fashion in the following, while short explanations for all 14 introduced metrics can be found in the Appx. B.

### 5.1.1 Confusion Matrix

Scientific evaluations of IIDSs base on labeled benchmarking datasets (cf. Sec. 4.2), including samples of cyberattacks (malicious) and benign behavior. After a training phase, for each data-point in the dataset, the known labels are compared to the

output of the IIDS (alarm or no alarm). The high-level goal of an IIDS is to detect as many attack instances as possible while emitting few (false) alarms for benign behavior. Note that especially in ICSs, where cyberattacks are rare compared to benign behavior, false alarms should be minimal [18].

As the first performance indicators, one can count the occurrences of all four possible combinations between dataset labels and IIDS outcomes called true-positive (TP), true-negative (TN), false-positive (FN), and false-positive (FP), making up the confusion matrix to capture an IIDSs behavior.

### 5.1.2 Point-based Metrics

Since there is a desire to express performance with a single value irrespective of the dataset, there exist a large variety of *point-based* metrics derived from the confusion matrix [61] (cf. Tab. 3). These express properties, such as its overall correctness (accuracy), the fraction of correct alarms (precision), or fraction of identified attacks (recall). Point-based metrics find wide application beyond IIDS research, e.g., machine learning, and thus a natural choice for comparisons.

### 5.1.3 Time Series-aware Metrics

Point-based metrics are suitable when the benchmarking datasets' entries are independent. However, ICSs are inherently time-dependent, i.e., the current state of an ICS is always a result of the system's previous state. Consequently, IIDS datasets extracted from these systems also need to be considered in the aspect of time, i.e., an alarm extending beyond an attack while the system did not yet reach its normal operational state should be interpreted differently from a false alarm in the middle of normal behavior. In such or similar scenarios, point-based metrics are skewed, which is already known in literature since 2014 by Gensler et al. [25].

Consequently, many novel *time-aware* metrics tackle such flaws [25, 32–34, 44, 69]. They, e.g., simply count the number of detected and continuous attack scenarios (detected scenarios) [48], aggregate the time it takes until the IIDS emits an alert after the attack began (detection delay), or define new time series-aware versions of precision and recall to favor early detection of an attack instance (e)TaPR [34]. Yet, Huet et al. [32] already found that (e)TaPR is not free of flaws and responded with their own Affiliation metric. Note that while time series-aware metrics like Numenta [44] or the one proposed by Tatbul et al. [69] and Gensler et al. [25] exist, they were observed only seldom in our SMS, if at all.

## 5.2 Metrics Utilized in IIDS Research

Given that a wide variety of metrics exist to express IIDS performance, in our final research question Q3, we ask how often and when these metrics are used. Overall in our SMSs, we found 167 different metrics and flavors, including, e.g.,

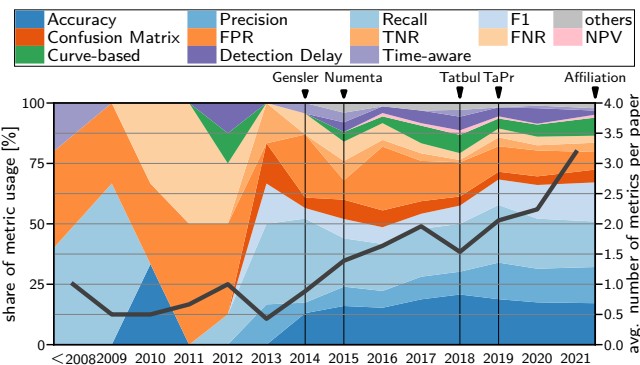

Figure 6: Point-based metrics dominate IIDS evaluations, with accuracy, precision, recall, and F1 being the top most used metrics. Over time, the number of metrics in a publication increased to currently 3.2 on average in 2021.

subtle deviations such as multi-class or weighted variants. To handle this amount of metrics, we aggregated them into similar classes, e.g., binary-class and multi-class accuracy are considered as the same metric type. Since a majority is used infrequently, i.e., only 12 occur at least ten times, we bundle rarer metrics into a single class (others) in the following.

### 5.2.1 Metrics over time

To obtain a first overview of the utilization of frequent metrics, we depict their use over time in Fig. 6. First of all, the number of different metrics used in a single publication on average (2.3 overall) kept increasing since 2013, and nowadays, publications use 3.2 metrics on average. This greatly coincides with the previous observation in Fig. 2, where the year 2013 marked the turning point when IIDS research took off. This trend toward more metrics contributes to higher comparability in the research domain and hints at in-depth evaluations. However, there also exist 157 publications that evaluate without any quantitative metrics and instead rely only on textual descriptions, e.g., elaborating which attack scenarios were detected or discussing results visually along graphs. Note that textual descriptions cannot be aggregated into a unified class as they differ significantly, i.e., two publications using textual descriptions hardly describe the same feature.

In contrast to dataset utilization (cf. Fig. 2), the metric utilization fluctuates less over time. One notable trend, again starting around 2013, is that accuracy, precision, recall, and F1, i.e., the classical point-based metrics, have established themselves as metrics with high usage by representing 63.1 % of all used metrics. At the same time, out of the 348 publications utilizing one of these four metrics, only 81 state all four. Thus their usage is inconsistent, and most publications only focus on certain aspects of their expressiveness.

Concerning all point-based metrics, which account for 93.3 % of all metrics, the confusion matrix resembles an

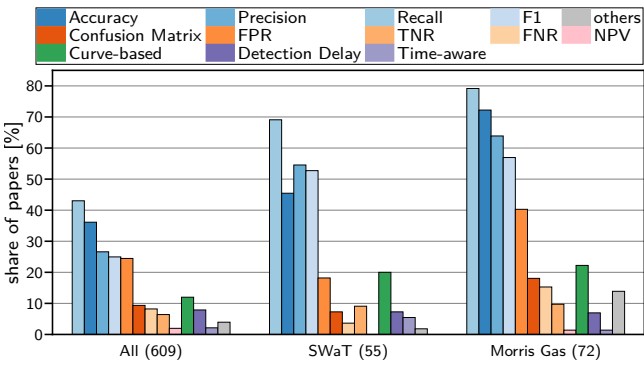

Figure 7: Papers utilizing SWaT and especially the Morris-Gas dataset are dominated by point-based metrics. Time series-aware metrics are slightly more frequent for SWaT.

important metric as it builds the foundation to calculate all point-based metrics (cf. Sec. 5.1.1). However, out of the 57 papers that publish the confusion matrix, just 19 fully state or discuss all four common metrics (accuracy, precision, recall, and F1), even though this would be easily doable. In 9.4 % of the publications where the confusion matrix is published, at least missing metrics can be calculated, which is not possible the other way round, i.e., the confusion matrix cannot be computed if, e.g., F1 scores are indicated. It thus remains questionable why publications omit frequently used metrics when all data to compute them has to be available anyway.

Even though it has been known since 2014 that for industrial IDSs, point-based metrics may be flawed [25], they make up 93.3 % of all metrics. As a time series-aware metric, detection delay receives constant but infrequent use by 48 publications overall. Still, detection delay alone does not quantify the portion of detected attacks and thus likely serves to enhance point-based metrics. Newer promising time series-aware metrics yet have to gain traction (only 13 publications use them), despite their added value in interpreting IIDS results.

Evaluations in the IIDSs research dominantly build upon point-based metrics, which are known to have flaws, especially on time-series datasets as used in IIDS research [25, 33].

### 5.2.2 Metric distribution on datasets

In Sec. 4.1.2, we observed the formation of obvious clusters in research around publications using the same dataset. Consequently, Fig. 7 depicts the dataset's influence on the chosen metrics. Therefore we pick the two most commonly used datasets, SWaT and Morris-Gas (cf. Sec. 4.2), and compare their metrics distribution against all publications.

The top four metrics (accuracy, precision, recall, and F1) play a major role for the SWaT and Morris-Gas datasets too, even more than across all publications. Recall, for example, is used in 43.0 % of all publications but indicated for 79.2 % of IIDSs evaluated on the Morris-Gas dataset. The order of usage between them is also similar, i.e., precision is used the

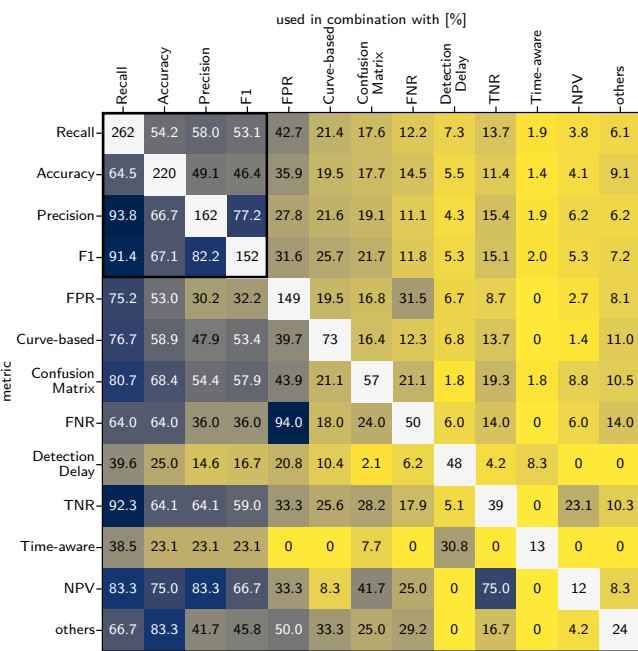

Figure 8: Metrics show strong correlations w.r.t. combinations they occur in publications. E.g., F1 is used in 152 publications, and among them 82.2 % publish precision. Vice versa, 77.2 % of the 162 papers with precision also state F1.

most and F1 the least. The only exception is accuracy, which is indicated less often for the SWaT dataset. This difference might be caused by SWaT featuring far fewer attack instances. Another exception is that other point-based metrics (confusion matrix, FPR, TNR, FNR, and NPV) receive greater attention in the Morris-Gas dataset. Contrary, time series-aware metrics are slightly more common for SWaT.

Our analysis highlights once again the dominance of point-based metrics, especially for the top two datasets by usage.

### 5.2.3 Metric Combinations

Even though there exists a variety of metrics (cf. Sec. 5.1), a single metric usually has to be considered in relation to others. E.g., precision and recall have to be discussed jointly since an IIDS which detects all attacks (high recall score) might do so simply by emitting alerts continuously, which would become visible in a low precision score. Fused metrics like F1 try to remedy this situation but deny in-depth reasoning afterward as they do not retain the precise original information. According to our SMS, publications state 2.3 metrics on average to sketch light on the IIDS performance from different perspectives. Consequently, as the last step, we evaluate which metrics are used together.

To this end, Fig. 8 depicts the occurrence of combinations between the considered metrics. On the diagonal, we enumerate how often each metric is utilized globally, i.e., recall is used 262 times. The remaining cells indicate how often the

indication of one metric leads to the usage of another metric.

In total, 152 publications used the F1, and 91.4 % of these papers (stating F1) also published recall values. This is not surprising since knowledge of the recall is required to calculate F1. Vice versa, however, 262 papers used recall, and of them only 53.1 % of those also published F1 scores. Looking at precision and recall as two complementing metrics, recall is used in 93.8 % of the publications that state precision. If recall is stated, only 58.0 % also publish precision. While the number of detected attacks (recall) is valuable information, for the 42 % of IIDS not indicating precision, it is unknown whether the IIDS indeed performs better than an IIDS that simply outputs one continuous alarm.

For popular point-based metrics (within the black rectangle), we observe a strong dependence between them, which is not surprising as these are heavily used (cf. Fig. 6 and Fig. 7). Since many point-based metrics are derived from the confusion matrix (cf. Tab. 3), the confusion matrix likewise has a high correlation with these four. However, it is not guaranteed that these are published reliably, as F1 is contained in only 57.9 % of the cases when the confusion matrix is presented. This is in line with our previous observation that of the 57 publications with a confusion matrix, only 19 state all of the four most often used point-based metrics (cf. Sec. 5.2.1).

Except for the dependencies between FNR and FPR, there exist few apparent correlations, thus often omitting the classical point-based metrics completely. Especially publications taking advantage of newer, time series-aware metrics lack other metrics. While this development makes sense (why should we indicate flawed metrics when we can use better ones), it makes comparisons to prior works harder.

## 5.3   Expressiveness of Metrics

Until now, our SoK on evaluations of IIDSs bases on theoretical observations from literature, e.g., which datasets and metrics are used. In the following, we extend our analysis beyond a literature mapping study with *practical* experiments to understand the quantitative impact of metric choices on the evaluation outcomes and to derive metrics that offer high expressiveness. To this end, we conduct a comparison study across ten IIDSs from research on two datasets and utilize our evaluation tool (cf. Availability Statement) to compare various metrics. Especially for newer time series-aware metrics, which are more difficult to compute [32, 34], no common library exists thus far. Besides the metrics discussed in the following, the tool provides a total of 18 point-based and 14 time-aware metrics, for which few implementations exist.

### 5.3.1   Experiment Design

As we observed in Sec. 4.2, the IIDS research community is governed by two major directions of datasets: network-based datasets such as the Morris-Gas [55] and process data

datasets such as SWaT [28] containing physical time series data. We aim to cover both types in our evaluation and thereby also cover two important IIDS types from research, namely knowledge- and behavior-based IIDSs (cf. Sec. 2.1). For knowledge-based IIDSs, we examine five supervised machine learning approaches [59, 72] originally evaluated on the Morris-Gas dataset. For behavior-based IIDSs training on process data, we leverage five anomaly detection approaches, with TABOR basing on timed automata [48], Seq2SeqNN utilizing neural networks [39], PASAD leveraging singular spectrum analysis [6], SIMPLE implementing minimalistic boundary checks [73], and Invariant mining invariant logical formulas [20]. Contrary to the supervised machine learning approaches on the Morris-Gas dataset, these IIDSs are evaluated on the temporally *ordered* SWaT dataset, which provides dedicated attack-free training data and testing data, including anomalies. As an interesting case for the SWaT dataset, we added an IIDS that randomly emits alerts by a 50 % chance.

### 5.3.2   Metrics Under Study

In this study, we focus on the four common point-based metrics accuracy, precision, recall, and F1 (cf. Sec. 5.2) and modern time series-aware variants of them called enhanced time series-aware recall (eTaPR) [34] (cf. Appx. B.2.3). More precisely, eTaP for precision, eTaR for recall, and eTaF1 for F1 (there is no time series-aware accuracy equivalent). Additionally, we consider the time-aware Affiliation metrics (again expressed as variations of precision, recall, and F1) proposed by Huet et al. [32], which claim to be robust against randomly generated alerts. These metrics, like their point-based counterparts, favor high detection rates but diminish the expressiveness of consecutive alarms if they start too early or overhang beyond the duration of an attack. Furthermore, we examine a variant of $F1$, which allows weighting precision and recall differently. This may be crucial in industries since cyberattacks are rare compared to normal behavior, preferring a high precision over false alarms. The datasets in our study already incorporate this class imbalance, with Morris-Gas containing 22 % malicious data, SWaT just 12 %, and real deployments likely observing even fewer attacks. Thus we examine $F_{0.1}$ in addition, weighting precision ten times more than recall. As the last metric, and since there is only one repetition for each attack type in SWaT, we discuss the percentage of detected scenarios (unique attack types).

### 5.3.3   Results

*Point-based.* We begin with analyzing the knowledge-based IIDSs on the Morris-Gas dataset in Fig. 9(a). Here, the point-based metrics (accuracy, precision, recall, and F1) coherently judge the IIDSs' performance, i.e., one IIDS is strictly better than another, and only in recall does the ordering between ExtraTrees and DecisionTrees flip. The $F_{0.1}$ variant's judg-

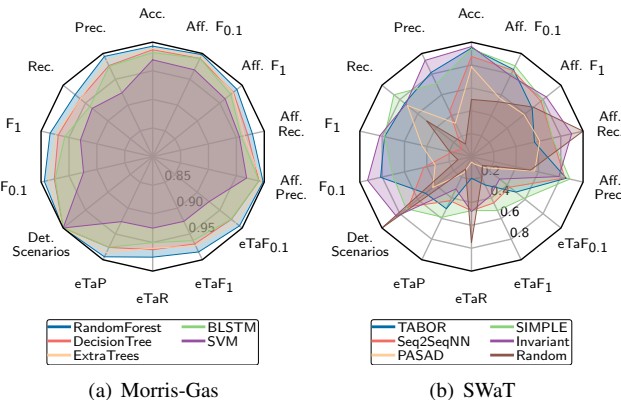

(a) Morris-Gas        (b) SWaT

Figure 9: While point-based metrics rate IIDSs' performance consistently on the Morris-Gas dataset (a), they fail to provide a coherent picture of the time-series dataset SWaT (b) and judge IIDSs better than time-aware metrics.

ment is in line with the other metrics, likely due to the high amount of malicious samples (22 %) in this dataset. Also, the time series-aware variants draw a nearly identical picture here. Note that the attack instances of the Morris-Gas dataset correspond to manipulations of individual network packets, and thus temporal effects are minimal. While the IIDSs are well at detecting these attacks, it is unclear whether the attacks themselves are actually comprehensible to ones observed on real deployments. Overall, the considered metrics coherently judge the IIDSs' performance on the Morris-Gas dataset.

The picture changes for the SWaT dataset comprising of time series of physical states (cf. Tab. 1). We additionally depict the raw alert emitted by the IIDSs over time in Fig. 10. First of all, as depicted in Fig. 9(b), all IIDSs perform well according to accuracy (more than 0.75). Yet, in comparison to all other metrics, accuracy seems to overestimate their capabilities. We attribute this to SWaT's composition comprising to 12 % of attacks (which is more realistic than Morris-Gas with 22 % as attacks are rare in practice), and an IIDS that emits no alarms at all would score an accuracy of 0.88 already.

Regarding precision and recall, we observe ambiguity. Seq2SeqNN falls far behind the other approaches in recall, which we attribute to a single long attack in SWaT (accounting for 63 % of all attack samples) being missed by the approach (cf. Fig. 10). Besides this attack, Seq2SeqNN achieves decent scores as it correctly detects most of the other attacks (cf. detected scenarios). Therefore, point-based metrics overvalue this attack, i.e., no obvious relation exists between the attack's duration and its severity that would justify this effect. In contrast to the Morris-Gas dataset, the $F_{0.1}$ score clearly favors IIDSs with higher precision, and thus TABOR is preferred over the SIMPLE IIDS (even though nearly equivalent in F1).

*Time series-aware.* In general, time series-aware metrics promise to solve these inaccuracies of point-based metrics. In our practical study, all IIDSs perform much worse on the

time eTa series-aware variants [34], which might be the case since they have not been designed for this kind of (potentially more valuable) evaluation. Here, the SIMPLE IIDS is now the best-performing approach according to eTaP, eTaF$_1$, and eTaF$_{0.1}$ as its emits alerts are precise, i.e., no overshooting as by PASAD or occasional short false-alarms as in TABOR and the Invariant IIDS. Yet contradicting the traditional recall score, Seq2SeqNN now belongs to the best IIDSs in the time-series recall pendant (eTaR) probably because the time-aware metric analyzes alarms consecutively, and thus false-negatives of overshooting alarms are not weighted that negatively.

The time-aware affiliation metrics [32] draw a completely different picture since all IIDSs perform much better. The ratings for precision, $F_1$, and $F_{0.1}$ are mostly consistent, and the IIDSs only differ significantly in terms of affiliation recall. However, a random IIDS, which the metric should consider as the minimum baseline [32], is counterintuitively perceived as a better approach than PASAD and TABOR in the affiliation $F_1$ score. In all other point-based and time-aware metrics, this random IIDS is perceived as the worst approach (except for detected scenarios and recall).

*False-positive resistance.* For practical deployment, IIDSs with many false positives are unsuitable [18], and thus identifying those in evaluations is crucial. In that regard, while the Invariant IIDS outperforms all other approaches in many metrics, visually (cf. Fig. 10) exhibits the least usable approach due to its plentiful but short-lived false alarms. Only in the eTa metrics it performs badly.

### 5.3.4 Conclusion

Point-based metrics draw a coherent picture for the Morris-Gas dataset containing a significant amount of attacks with few temporal effects as single network packets were manipulated. In contrast, authors have to carefully examine their results on the SWaT dataset since, depending on the chosen metric, their IIDS may perform excellently or poorly. These results are in line with Fung et al. [23], finding that time-series metrics are preferable for reconstruction-based IIDSs and point-based scores may be misleading. For the affiliation metrics by Huet et al. [32], our experiment challenges their results, especially for an IIDS that emits alerts randomly. Thus, a better understanding of how such newer time series-aware metrics have to be interpreted is crucial.

Overall, it is unlikely that a single metric exists that catches all industrial operators' different goals, e.g., preferring few false alarms over detected attacks. IIDSs should be evaluated with different metrics to truly highlight their capabilities, as cherry-picking metrics may lead to misleading results. The $F_{0.1}$ score provides an interesting alternative for more realistic scenarios. Furthermore, visual comparisons exhibit a non-negligible added value to evaluations too. Lastly, the knowledge- and behavior-based IIDSs are hardly comparable today since they are divided by dataset type.

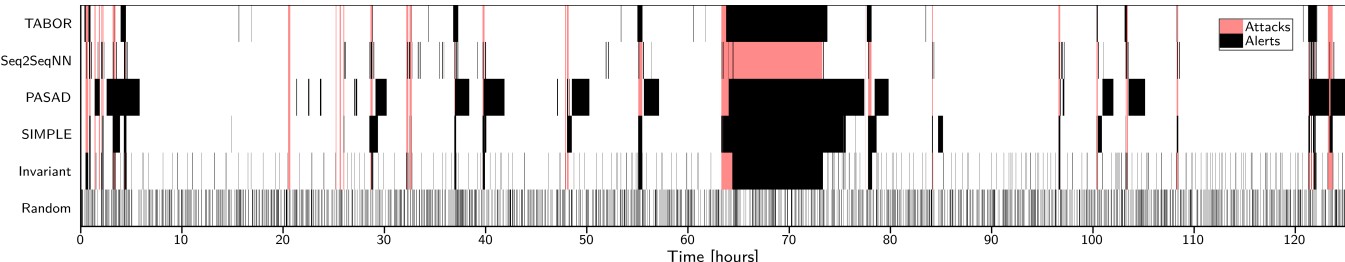

Figure 10: Visualizing alerts side-by-side provides an in-depth view of their distinct alerting behavior. E.g., the underwhelming performance of the Seq2SeqNN IIDS in point-based recall (cf. Fig. 9) can easily be attributed to a single prolonged attack of the SWaT dataset. Note that alerts have been extended to a minimum width of 1 minute for visibility (except for Random).

# 6 Common Issues & Recommendations

The huge potential of IIDSs to combat rising threats from cyberattacks against industrial networks is indisputable. Unsurprisingly, our systematic analysis (Sec. 4 and Sec. 5) shows an unbroken and increasing interest in this research field (40.9 % average yearly increase between 2013 and 2021), with at least 609 publications investing great efforts in proposing IIDSs, which are complemented by further work on creating datasets, designing evaluation metrics as well as surveys and meta-analysis. However, our SMS also unveils and quantifies flaws in this field that hamper scientific progress. Thus, in the following, we synthesize common issues persisting in IIDS evaluations and distill recommendations to move forward to more thorough IIDS evaluations.

## 6.1 Common Issues in IIDS Evaluations

Our systematic analysis of the IIDS research field reveals that the current state-of-the-art w.r.t. evaluation methodologies has serious inefficiencies, eventually slowing down the overall progress in securing industrial deployments. Our SMS, covering the body of literature until 2021, enables quantifying these inefficiencies and makes (promising) trends visible in contrast to previous meta-surveys and experiments on a usually narrower scale (cf. Sec. 2.3). More precisely, we identify three issues (I1–I3) prevalent in evaluations of IIDSs and present them along the results from our SMS in the following.

▶ **I1: Dataset Diversity.** *We identify a lack of diversity in datasets used for evaluations.* Regarding the utilization of datasets, we find that IIDSs are evaluated on 1.3 datasets on average (cf. Fig. 4), which aligns with 1.32 datasets on average reported by related work [74]. Notably, the majority (501 publications) considers only a single dataset, despite a significant selection of datasets being publicly available (we identified 35 public datasets in our SMS, and other work lists 23 datasets or 61 industrial testbeds [14]). Since there exists this large gap between available and utilized datasets, this raises the question of why many datasets are only used rarely. Possible reasons include datasets being too narrow

in scope (e.g., focusing on single attack types), too small (providing only few training or testing samples), difficult to use (e.g., requiring in-depth knowledge of a specific industrial protocol), or simply not widely known among researchers. Lua et al. [49] also find that high-quality datasets are rare. Moreover, for the few publications that evaluate multiple datasets (16.4 %), these datasets mostly stem from the same origin (cf. Tab. 2). Thus, IIDSs' evaluations are mostly confined to a single scenario (dataset) and do neither cover the diversity of industrial domains nor communication protocols (cf. Sec. 4.2.2). Consequently, it remains unclear whether IIDSs are applicable outside the narrow scenario they have been evaluated in, making real-world deployments risky and requiring repeated efforts for different scenarios.

▶ **I2: Metrics Ambiguity.** *Metrics used in evaluations and comparisons pose ambiguity regarding the actual detection performance of IIDSs.* Due to the unclear and biased choice of metrics, the *actual* detection performance of proposed IIDSs often remains unclear, as also claimed by Giraldo et al. [27]. Seemingly promising, we observed an increase in the number of utilized metrics (3.2 per publication on average in 2021) while simultaneously moving away from mere textual descriptions (cf. Sec. 5.2) toward established point-based metrics (cf. Fig. 6). Accuracy, precision, recall, and F1 make up the majority of utilized metrics again [49]. However, we also encountered a total of 167 flavors of metrics, e.g., subtle variations such as multi-class or weighted scores, which further complicates metric ambiguity. At the same time, essential metrics, expected to be provided in combination, are often omitted or incomplete in publications. I.e., of the 57 publications providing the confusion matrix, only 19 state accuracy, precision, recall, and F1 in combination (cf. Sec. 5.2.1). Even more severe, precisely these four point-based metrics, making up 63.1 % of the metric usage, do not accurately capture the detection performance in time-series scenarios [23] and are skewed towards the detection of long-lasting attacks (cf. Sec. 5.3). While plenty new metrics [25, 32–34, 44, 69] are designed that supposedly address these issues, these metrics are rarely used in evaluations (only 13 publications), likely because a broad understanding about their expressiveness is

missing. Lastly, as our practical experiments show, not a single metric can describe all aspects of an IIDS, and visual comparisons can disprove, e.g., seeming promising IIDSs.

▶ **I3: Underutilized Comparability.** *Evaluations of IIDSs do not capitalize on the large potential for comparisons among the vast body of existing research.* The number of comparisons to related work performed by new IIDSs' has experienced earlier criticism already [27]. On average, an IIDS is only compared with 0.5 other proposals, slightly more than observed in previous works (0.38) [74]. Yet, in theory, authors could compare an IIDS to an average of 6.0 other approaches sharing at least one common dataset and metric (cf. Fig. 5). Simultaneously, the current state of the research field leaves researchers large freedom to choose from any of the theoretically suitable publications for their comparisons. This situation is even aggravated by the sparse commitment to publish artifacts (cf. Sec. 4.3.1), which leaves researchers no choice other than to reproduce others' works, e.g., to ultimately conduct comparability studies—a non-trivial task that is prone to failure [17]. Meanwhile, researchers have to rely on public datasets and the expressiveness of metrics that both exhibit flaws themselves (cf. I1 and I2). However, proper comparisons are essential to better understand if and how a novel IIDS improves upon existing work and thus collectively move the research field forward.

## 6.2 Recommendations for IIDS Evaluations

To address these prevalent issues and thus enhance evaluations as well as the applicability of future IIDSs research, we extract key aspects from our systematic analysis and turn them into six actionable and practical recommendations (R1–R6).

Since our recommendations target different parties involved in IIDS research, we address them to (i) *researchers* designing new detection approaches, evaluating them, and comparing them to the state-of-the-art; (ii) dataset *creators* recording qualitative datasets or providing simulations and testbeds; and (iii) industrial *operators* with precise knowledge of the individual needs of ICSs' striving to role out IIDSs in practice.

▶ **R1: Evaluate More and Diverse Datasets.** *Researchers* should use the many readily available datasets to comprehensively evaluate their IIDSs for different industrial domains, communication protocols, and attack types. Using multiple, especially diverse datasets avoids overfitting [73], boosts generalizability across ICS, enables insights across multiple domains, and allows assessing the potential efforts required to facilitate (widespread) deployability across industries. For a concise dataset selection, we recommend focusing on publicly available datasets such as those listed in Conti et al.'s [14] comprehensive datasets and testbeds overview. For evaluations requiring process data, datasets of multiple origins and industrial domains should be used. Likewise, for IIDSs operating on network traffic, generalizing the approach to different industrial protocols should be considered. More-

over, specialized datasets that, e.g., model a single attack type, cover a niche industrial domain, or deploy rarely used protocol, still provide substantial added value when used in combination with other, more general datasets to better understand the capabilities and limitations of an IIDS. Additionally, researchers can consider datasets containing attacks and faults (e.g., the IEC61850SecurityDataset [12]) to evaluate whether their proposed IIDSs can differentiate these kinds of unwanted behavior to facilitate swift and correct reactions by operators to alerts. Lastly, to ease evaluations on a multitude of datasets with potentially varying formats, agreeing on unified dataformats, such as IPAL [74], may help lower the burdens for researchers.

▶ **R2: Provide High Quality Datasets.** Dataset *creators* should provide the research community with high-quality and diverse datasets to counteract the current bias to two major datasets (cf. Tab. 1). To ensure the practical relevance of datasets, they should ideally be generated in close cooperation with industrial partners [57] since otherwise, IIDSs designed upon them risk not being of practical use to industrial *operators*. Such collaborations, even though costly [7], also allow enriching datasets with properties and demands of actual industrial deployments, e.g., the criticality of an attack, an acceptable delay until which a detection is excepted, or documentation of how long the ICS behaves abnormally after an attack until it stabilized again. Furthermore, research lacks datasets that tackle the needs of all IIDS flavors (cf. Sec. 4.2.2), inhibiting a consolidation of the overall research landscape. For one, only a few datasets (Faramondi et al. providing a rare exception [19]) combine network traffic and process data, which is necessary to compare IIDSs that work on these different data types. Moreover, datasets should be designed and created such that they are applicable to both supervised and anomaly-based IIDS training (currently, no corresponding dataset is known to us), e.g., by including repetitions and variations of the same attack, providing sufficient long samples of benign behavior, and including novel attacks, which are not previously trained on, to avoid the drawing of false conclusions [43]. For more concrete advice on how scientific IIDS evaluation datasets should be designed, please refer to the works by Gómez et al. [58] and Mitseva et al. [54].

▶ **R3: Use Standardized and Accessible Metrics.** *Researchers* should carefully consider the use of metrics and rely on both common (flawed) metrics for comparability as well as recent time-series aware metrics (cf. Sec. 5.2) that attempt to mitigate known flaws. In that regard, meta-studies on how metrics fare against each other, as done in Sec. 5.3 and by Huet et al. [32], help understand the expressiveness of evaluations. Ideally, a wide variety of different metrics is used to disseminate the performance of newly proposed IIDS, which would also facilitate comparisons in the future. Especially with the rise of new metrics and to standardize the evaluation process, *researchers* should be equipped with adequate tooling to calculate these metrics easily. Our evalua-

tion tool used in Sec. 5.3 and published along this paper will greatly help in that regard. To facilitate a sensible choice of metrics and ensure comparability of related IIDS approaches, dataset *creators* should explicitly define standard evaluation metrics for their datasets, as has been done, e.g., for the HAI dataset [65]. First, fixing metrics a priori ensures the neutrality of evaluations and reduces potential biases in their selection by researchers. More importantly, however, dataset developers know the underlying ICS best, e.g., w.r.t. the impact of false positives or the likelihood of attacks. Often they are the only people with the necessary expertise to identify the demands of a cybersecurity solution and, thus, the most valuable metrics to benchmark an IIDSs in their scenario.

▶ **R4: Facilitate Comparability With Public Artifacts.** *Researchers* should make the artifacts publicly available [18], especially IIDS implementations, underlying their work to facilitate comparability of IIDS research. If artifacts cannot be provided, e.g., due to licensing issues or private datasets, we recommend that researchers at least release the precise IIDS outputs, e.g., a list with all packets classified as malicious by an IIDS. These outputs, together with the (anonymized) labels of the dataset, suffice to calculate new metrics retrospectively, thus gaining new insights into the IIDS's performance even after publication. Furthermore, publishing an IIDS's alerts when evaluated on a public dataset is also valuable if published alongside its implementation, as getting research code to run and produce the same results independently is often hard work (e.g., due to lacking documentation), especially some years down the road. Such published labels directly avoid the current lock-in to metrics during the time of publication, thus greatly enhancing the comparability of IIDS research. This freedom is especially crucial in an early stages of IIDS research since it is unknown which metrics and evaluation methodologies will eventually gain acceptance.

▶ **R5: Strive for Continuous Feedback Loops.** *All* stakeholders should strive for coherence and applicability of IIDS research. *Researchers* should avoid proposing isolated IIDSs without proving their necessity and bridge the gaps between related branches for greater coherence [74]. At the same time, meta-surveys that critically review the state-of-the-art have to provide directions regarding which approaches work well in given settings, which datasets and metrics are suitable, and which approaches should IIDSs should compare to. Lastly, a continuous exchange between all stakeholders should be established [57], e.g., in the form of public talks, workshops, or the dissemination of scientific publications. Only then can industrial *operators* stay informed about recent advancements and likewise keep dataset *creators* updated to ensure overall research strives for practical applicability. As an initial step in that direction, we provide the artifacts of our broad SMS, which can serve as the foundation for future surveys on more specific topics, such as in-depth analyses of the proposed detection methodologies or benefits and drawbacks of the wide variety of (newly proposed) evaluation metrics.

▶ **R6: Think Beyond Alerting.** *Researchers* should extend their focus beyond optimal attack detection coverage and the required actions after IIDS alerts. Such actions may include steps to understand the alert [18, 64], localize the attacker [4], mitigate an attack's damage potential [67], recover the system to a safe state [71], and lastly, perform forensics to learn for the future [37]. Given this chain of tasks operators have to execute, which may include temporal interruptions of the process, it may also be crucial for researchers to consider the costs of (false) alarms emitted by their solutions. While research on follow-up procedure of IIDS alerts is currently critically underrepresented in the literature, this is partially caused by the secrecy of industrial *operators*. The sharing of detailed information about the operation of real-world ICSs allows researchers to propose valuable and actionable improvements to current processes. Moreover, this information also allows researchers to design suitable evaluation methodologies to evaluate the performance of the processes following an alarm. Overall, IIDS should thus no longer be considered as an isolated system, but the step from detection to (incident) response should be considered a tightly interlocked process.

## 7 Conclusion

The ongoing digitization of industries and increasing exposure of ICS to the Internet are accompanied by a rise in cyberattacks. Consequently, the new research field of industrial intrusion detection, promising to provide an easily deployable solution to uncover even sophisticated attacks, gained traction. In 2021 alone, 130 new detection approaches were proposed.

This SoK presents the first systematic attempt to shed light on this fast-growing research field and how different approaches are evaluated. Our thorough analysis of 609 publications reveals the tremendous efforts invested by the community to protect industrial systems. However, when it comes to evaluating detection approaches, we uncover widespread issues w.r.t. dataset diversity, the ambiguity of metrics, and missed opportunities for comparability, hampering the overall progress of this quickly growing research field. Based on our systematic analysis, we formulate actionable recommendations to overcome these issues and thus bring the entire research domain forward to sustainably and significantly improve the security of (real-world) industrial deployments.

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

# A  Systematic Literature Review

To find relevant literature proposing IIDSs with the help of various search engines, we derived a search string during the design of the SMS (cf. Sec. 3). As depicted in Fig. 1, the search string combines collections of keywords for the phrases *industrial* and *detection*. After validating the outcome of several search strings and keywords against known literature of that research landscape, we derived the following search string utilized in the SMS:

*("Industrial Control Systems" OR "Process Control Systems" OR "Supervisory Control and Data Acquisition" OR "PCS" OR "ICS" OR "SCADA")*
*AND*
*("Attack Detection" OR "Anomaly Detection" OR "Intrusion Detection")*

The final search string was applied to the combination of titles, abstracts, and keyword of all publications. Note that search engines do not care about capitalization.

# B  Overview on Utilized Metrics

Detection performance metrics quantify the capabilities of an IIDS, i.e., its ability to differentiate benign from malicious behavior. Thus, such metrics are essential to achieve objective comparisons among publications in research. Aside from subjective textual descriptions, we found a large variety of different metrics and flavors throughout our SMS, ranging from well-defined and widely adopted ones to novel proposals. To facilitate a systematic overview, we have derived a taxonomy in Sec. 5.1. In the following, we now provide additional details about the most important metrics, covering

well-established point-based metrics (Sec. B.1) and promising time series-aware metrics (Sec. B.2). Please refer to Tab. 3 for synonyms by which these metrics are also known.

## B.1   Point-based Metrics

The "traditional" way to evaluate IIDSs is to utilize a benchmarking dataset which includes a label for each entry in the dataset, stating whether this entry is benign or malicious, i.e., it corresponds to a (specific) cyberattack. Note that multiclass IIDSs and corresponding metrics also exist, e.g., IIDSs that precisely identify and attribute each conducted cyberattack type. However, for the sake of simplicity, we only refer to binary (malicious and benign) metrics in the following.

Given the labels of a dataset and the outputs/alerts of an IIDS, one can compare them point-to-point to estimate in how many instances the output is correct ($T$) and how often it is false ($F$), i.e., deviating from the expected label. Various metrics then derive performance scores with different meanings, usually normalization in the interval of $[0,1]$. For a detailed discussion on each metric, please refer to [8, 61].

### B.1.1   Confusion Matrix

Beginning with the dataset's labels and the IIDS's alarms, four different outcomes are possible: First, true negatives ($TN$) are all instances where the dataset label is benign, and the IIDS has not raised an alarm. Likewise, true positives ($TP$) correspond to those attack instances within the dataset that are correctly identified as attacks. In contrast, false negatives ($FN$) are attack instances that are incorrectly not detected by the IIDS. Lastly, false positives ($FP$) are false alarms triggered even though no attack has occurred.

Counting all of these four possible outcomes across a dataset yields the confusion matrix, laying the foundation for many point-based metrics introduced in the following.

### B.1.2   Recall / True Positive Rate (TPR)

This metric states how many attacks of the dataset are actually detected by an IIDS. Naturally, an IIDS has to detect as many attack instances of a dataset as possible.

$$\frac{TP}{TP+FN}$$

### B.1.3   Miss Rate / False Negative Rate (FNR)

In contrast to TPR, FNR measures the fraction of missed attacks. Hence, a lower score is preferred.

$$\frac{FN}{TP+FN}$$

### B.1.4   Specificity / True Negative Rate (TNR)

Since cyberattacks are rare, it is crucial that an IIDS does not trigger alarms during benign system behavior. Thus, TNR defines the fraction of correctly classified benign behavior. A high TNR score is preferential.

$$\frac{TN}{TN+FP}$$

### B.1.5   Fall-out / False Positive Rate (FPR)

Similar to TNR, IIDSs should only trigger an alarm in case of actual attacks. Therefore, FPR calculates the fraction of false alarms across the dataset, which has to be as low as possible.

$$\frac{FP}{FP+TN}$$

### B.1.6   Precision / Positive Prediction Value (PPV)

When focusing on the alarms triggered by an IIDS, PPV defines the fraction of correctly detected attacks among all existing attacks in the dataset.

$$\frac{TP}{TP+FP}$$

### B.1.7   Negative Prediction Value (NPV)

Contradicting the PPV metric, NPV counts the number of correctly classified negative predictions among all attack free parts in the dataset.

$$\frac{TN}{TN+FN}$$

### B.1.8   Accuracy

The first metric capturing the overall number of correct classifications is accuracy. The higher the accuracy score is, the more reliable the predictions of the IIDS are.

$$\frac{TP+TN}{TP+TN+FP+FN}$$

### B.1.9   F1

For intrusion detection, there is an inherent tradeoff between achieving a maximal number of detected attacks (TPR) while reducing false positives (expressed by PPV as correct alarms). The F1 score combines both design goals into a single metric through the harmonic mean. Note that F scores with different TPR and PPV weightings exist, as discussed in Sec. 5.3.

$$\frac{2TP}{2TP+FP+FN}$$

### B.1.10    Receiver operating Characteristics Curve (RoC)

IIDSs and their detection models may require fine-tuning hyperparameters, e.g., to determine a threshold upon which an alert is triggered. Since the previous metrics evaluate IIDSs for a fixed setting, it is impossible to describe their behavior across the parameter range.

The RoC curve is a method to visualize multiple IIDS configurations and their performance by plotting FPR on the x-axis and TPR on the y-axis. Since each entry represents a tradeoff for a specific IIDS model, the RoC curve enables developers to choose a suitable configuration visually. For a detailed discussion on the appropriate usage of RoC curves, please refer to Arp et al. [8].

### B.1.11    Area under Curve (AuC)

The Area under (the RoC) Curve abstracts from a visual performance indicator and defines a quantitative metric expressing IIDS performance for a variety of configurations by integrating the enclosed area. If only a single configuration is measured with the RoC visualization, AuC can be simplified into to the following formula [61].

$$1 - \frac{FPR + FNR}{2}$$

## B.2    Time Series-aware Metrics

Besides point-to-point comparisons between dataset labels and IIDS alarms, recent metrics strive towards time series-awareness [24, 33–35, 38, 44, 69]. They usually define attacks and alarms as a continuous time range with start and end points. Alarm intervals should be largely overlapping, i.e., an alarm is expected immediately after the start of an attack and should stop in time after the attack phase.

### B.2.1    Detected Scenarios

In contrast to point-based metrics and since time series-aware attacks are considered a single instance, it suffices to be indicated by an IIDS with a single short alarm. Unlike point-based metrics, attacks are considered a single instance in the time series-aware domain, and thus it is sufficient for an IIDS to trigger a single alarm. Therefore, detected scenarios enumerates the number of independent attack instances detected by at least a single alarm.

### B.2.2    Detection Delay

Nonetheless, early detection is still preferential in time-critical scenarios as this increases the time to respond to an attack. Thus, for all attacks in the dataset, the detection delay aggregates the time intervals between the start of an attack and the time of the first detection.

### B.2.3    Enhanced Time-aware Precision and Recall

Recently in 2022, Hwang et al. [34] proposed their (enhanced) time series-aware variants for classical point-based metrics, i.e., precision, recall, and F1, addressing known issues when adopting point-based metrics to time series-aware evaluations. For instance, while point-based recall weights long attacks as more important, the new time series-aware recall variant (eTaR) treats all consecutive attacks equally. To replace precision, eTaP implements diminishing returns for long-lasting alarms. Lastly, the new proposed eTaF score is defined in the same way as the regular F score (cf. Sec. B.1.9) but leverages the substitute eTaP and eTaR metrics.

### B.2.4    Affiliation Metrics

A similar approach was taken by Huet et al. [32] with their affiliation metrics. Again they consider alerts and the ground truth as continuous time-ranges instead of independent points. In the first step, their approach associates each alert to the closest ground truth, called local affiliation, and then calculates the individual distances for precision and recall. What makes their approach interesting is, that the final result is normalized in comparison to an IIDS that emits alerts at random. As also done for eTaF1, the time-aware affiliation precision and recall variants are averaged into the affiliation F score.

