# OpenReview forum: "SoK: Evaluations in Industrial Intrusion Detection Research"
_JSYS/2023/March_Papers — Revise_

### Official Review · Reviewer_JHfT · 2023-04-15

**Decision:**

Strong reject: this paper has serious problems, fixing it would definitely take more than three months

**Strengths:**

The paper is extremely well written. I also like the fact that most claims, even the simple ones, are backed by citations. I particularly enjoyed reading the IIDS landscape classification in Section 2.1.

Section 5.3 presents a set of practical experiment conducted to evaluate different IIDSs on two public datasets. The results allow a better understanding of the evaluation metrics across these datasets and using existing IIDSs.

Section 6 nicely summarizes the common issues and recommendations from the survey, with detailed discussion around each point.

**Weaknesses:**

One of the main concerns highlighted frequently in the paper is the lack of uniformity, either in terms of datasets or metrics or comparable experiments, across IIDS research. However, my concern is that if IIDSs are designed for different industries, which can vary significantly in terms of both size and form and function, is the need for uniformity and the need for comparison even justified in the first place? I would suggest addressing this concern in the beginning. The paper must convince the reader that the properties suggested are desired despite the diverse nature of IIDSs in practice.

Despite the focus of this paper on evaluation and metrics, I think it is still important to explain at least one case study on IIDS and the resulting dataset in detail to the reader. Currently, the paper does not contain any of it. It assumes that the reader is familiar with at least the main ideas in IIDS research, which may not be the case.

I didn’t quite understand at first what is a Systematic Mapping Study. Unlike a SoK, a SMS is not a familiar term.

Overall, feel more like a Survey paper with lots of quantitate insights than qualitative insights. The paper is not written keeping in mind an audience who does not work on IIDS.

**Detailed Comments:**

Dear authors, thank you for submitting your paper to JSys!

My most important concern is that the paper feels more like a survey paper than like an SoK paper. The JSys website says the following about an SoK paper: "Imagine that the SoK paper will be given to a junior PhD student. After reading it, they should have a good idea of where the area stands, and what challenges they could pursue. A great SoK paper will help set the agenda for the field by identifying research problems that could be worked on for the next 5 years. We do not require this of all SoK papers, but it is the ideal to aim for.” The current manuscript falls significantly short of an ideal SoK paper.

For instance, while I think the IIDS problem is well motivated throughout the paper, I think Section 4.1.1. is possibly redundant. Similarly, while Section 4.1.2 would have been useful in assessing a survey paper, I think for an SoK paper, where readers may not care so much about the papers surveyed but more about the contents and the topics covered, the section seems redundant. Along similar lines, Figure 4, for instance, is useful in a survey paper, but it is not useful at all in an SoK paper. I think the problem is the because the topic of this paper is *evaluations* of IIDS, it ends up exploring quantitative questions, which are then answered using statistics. However, this need not be the case. The paper can instead focus on a couple of case studies and explain the resulting datasets in detail (this could be Morris-Gas and SWaT), and then spend the majority of sections on evaluating and discussing different IIDSs and metrics on these and qualitatively explaining the results. Hence, I believe Sections 5.3 and 6 should be the main focus of SoK. Everything else seems like a survey paper.

Some clarification questions in the introduction and the second section:

“But, retrospectively integrating these measures into existing ICSs, operating for decades, is costly, if possible at all, due to their strict requirements toward, e.g., availability and latency.“ Is the problem that integrating security protocols like TLS onto ICS networking protocols like ModbusTCM requires more bandwidth and hence affects the latency? How is availability impacted? Irrespective to the answers to these questions, please elaborate on the reasons.

“Thus, the attacker model determines an IIDS’s input data, …” Does this assume that programs (i.e., code or application logic) are always correct, or can an attacker also affect code integrity?

“While knowledge-based systems …, behaviour-based IIDSs …” From this definition, both knowledge- and behaviour-based IIDSs sound similar. How are known patterns different from normal behaviour?

“process-based detection can leverage …” What do you mean by process-based detection? Are you referring to physical processes inside an operating system or inside the industrial plant? What is a critical state?

“Moreover, the IIDS’ performance needs to be measured based on sensitive metrics.” Which metrics are sensitive and why?

Can you explain in short, in a footnote perhaps, what is a Systematic Mapping Study and what does it entail?

**Expertise:**

Published in this area in the last 5 years

**Summary Of Review:**

Paper summary: The paper presents an SoK on research on Industrial Intrusion Detection Systems (IIDS), specifically, on the evaluation methodology used in IIDS research. The SoK relies on knowledge extracted from a large set of systematically surveyed papers on this topic. The SoK is centred around three questions. First, what types of datasets (public or private, network- or process-based) and how many datasets are used in IIDS research typically? Second, is IIDs research reproducible and on a related note are results from IIDS papers comparable? Third, which metrics are commonly used to evaluate IIDS? To answer the third question, the authors also present results from a practical experiment where they evaluate a range of knowledge- and behaviour-based IIDSs using the Morris-Gas (a network-based dataset) and SWaT (a process dataset) datasets. The SoK concludes with a discussion highlighting the common issues in IIDS evaluations in the recent years, and recommendations for IIDS evaluations in future.

The paper is very well written, but my decision is primarily based on the fact that the submission is more a survey than an SoK. In its current form, it is difficult for readers not already working on the specific problem of IIDS to gain anything from the paper, where as I believe that is one of the primary goals of an SoK (i.e., to educate a slightly broader audience). I do find that the topic in itself is useful for the larger systems/CPS/security community.

**Useful:**

yes

---

### Official Review · Reviewer_Rhap · 2023-04-28

**Decision:**

Weak accept: good paper with flaws that can be fixed in three months

**Strengths:**

Given below is the evaluation of the paper on the aspects expected from an SoK paper.

+ Qualitative analysis of existing research

  The paper does a good job of scanning and quantifying the evaluation metrics for IIDS. Using keyword search together with the transparent criterion for selection, this work "narrows down" the scope of IIDS evaluation papers to ~600. The rationale for systematizing aspects of IIDS are clearly motivated. This paper then investigates the datasets used in IIDS literature, the coherence among the literature, types of evaluations, and the comparability of different techniques.

+ Presenting a convincing, comprehensive taxonomy

  This is where the paper really shines. There is a very detailed evaluation and taxonomy of evaluation metrics of IIDS papers. A significant contribution of this work is in highlighting the confusion matrix on all the published works on such evaluation metrics. Given that intrusion detection algorithms do evaluate the true positive, false positive, false negative, and true negative values on data sets; it is natural to calculate all the evaluation metrics for all of the intrusion detection systems. The debate of whether a given metric is relevant to a given domain can be left to the discretion of the domain experts. Nontheless, I now can empathize with the frustrations of the authors on the discussion on evaluation metrics.

  In addition to the confusion matrix, this SoK classifies the IIDS research into different classes based on the datasets used and the metrics used.

+ Identifying future research directions or challenges that require the community attention

  This is where the paper is weak. The issues highlighted are not very surprising (not necessarily a bad thing) and the recommendations too seems standard (again, not a bad thing). I would have preferred some more creativity in the recommendations. Additionally, the authors could have highlighted the important metrics of interest in different domains (at least for the most widely used dataset) and continued some discussion around it.


**Weaknesses:**

Drawbacks

- I feel that the authors might have narrowed down the scope of the work too much by exclusively focusing on the evaluation metrics. While it gives a very clear target for scoping the work, a graduate student going through this work would only focus on the evaluation of IIDS. The student would still be unaware of the major techniques used in IIDS (or even some of the prominent works in this domain). I understand that the title itself mentions *evaluations*, however, I think purely focusing on evaluations without any discussion on the techniques is probably lacking.
- I have a concrete recommendation on the organization, I think the section on Reproducibility and Comparibility would be useful if it was discussed after the survey on evaluation metrics. A junior graduate student might not be aware of all the various metrics on evaluation and hence the reproducibility would make more coherent organization. It also fits very well with the recommendations that the authors propose.


**Detailed Comments:**

I appreciate the effort in performing this systematization. I think the suggestions can be incorporated into the paper in three months and the scope of the paper can be improved more.

**Expertise:**

Please contact the Area Chair if your expertise is lower than this

**Summary Of Review:**

This paper systematizes the research on industrial intrusion detection systems (IIDS) and primarily focuses on the evaluation of such industrial intrusion detection systems. I see a major drawback in the scope of the work; primarily, purely focussing on the evaluation metrics without any ties to the technique used. I think sometimes the evaluation metrics can be strongly correlated to the technique used. Furthermore, it might be important to classify the works based on the explainability or other aspects of the evaluation. Focusing on the evaluation without any reference to any of the techniques used for evaluation, could be considered lacking.


**Useful:**

yes

---

### Official Review · Reviewer_q5nt · 2023-05-02

**Decision:**

Weak reject: interesting papers with flaws, not sure if they can be fixed in three months

**Strengths:**

- Quite an extensive study with many metrics used to compare across the vast landscape of IIDS-focused research.
- After reading the paper, the issues pointed out by the reviewers in Section 6.1 follow pretty naturally.
- Most of the recommendations made by the authors are spot-on, given the highlighted issues.

**Weaknesses:**

- I am not convinced that the authors achieved the goal of performing a systemization of knowledge (SoK).
- The paper is deprived of technical details about the challenges and key techniques that represent the pillars of the field.
- Some recommendations are less grounded in the presented data and are more reflective of the opinions of the authors.

**Detailed Comments:**

First and foremost, I commend the authors for the impressive width of their analysis. The fact that the initial selection of papers included more than 900 works and was then narrowed to 609 publications is nothing short of remarkable.

This reviewer is only marginally familiar with IIDS. While I am fully aware of their importance and the fast-paced expansion of the field, I am not too grounded on the key techniques that have propelled the field forward in the recent past.

With this in mind, my expectation from an SoK paper was to "catch up" with the latest and greatest advancements in the field of IIDS and understand what is currently considered the open challenges and promising future research directions. Instead, the paper conducts a meta-analysis of the field. In other words, the manuscript reads a lot like a "state of the union" address to researchers *already familiar* with the area, highlighting the issues with the trends in the evaluation methodology employed in the surveyed works.

The abovementioned issue is the most significant weakness in an otherwise excellently written paper. In many ways, I am divided in providing a recommendation for this paper. I can see the value for the IIDS community to have the study presented in this paper published. But at the same time, this work is just a subpar SoK and more of a "call for action" type of paper.

In terms of final takeaways, the manuscript mostly delivers on its promises. The authors identify three main issues. First, the issue with dataset diversity (I1) emerges from the provided data on the number of datasets used by the considered papers. Next, the insufficient degree of comparison between newly published and previous works (I3) is also something easy to agree on, given the current (0.5) vs. potential (6.0) number of solutions each article compares against. Third, the issue with metrics ambiguity is perhaps less of a problem since authors who want to compare against related work can often bridge the gap in the notation/definition of the considered metrics.

Most of the recommendations made by the authors follow directly from the I1-I3 issues identified in Section 6.1. This is the case for R1-R4. Nonetheless, R5 and R6 are comparatively less obviously grounded in the presented data and read as more subjective recommendations based on what the authors would like to see in the near future.

## Final Recommendation

In its current shape, the paper is not a good fit for an SoK study. At the same time, the paper is well written. To align it better towards an SoK, the authors could complement the manuscript with additional sections that provide an actual assessment of the recent advancements in the field of IIDS by building on top of what the cited surveys [49] and [75] already do. Therefore, my recommendation is for a major revision. The focus of the revision should precisely be on delivering an understanding of the state of the art (not just the trends in evaluation methodologies) for the uninitiated reader in IIDS. I am confident that the hybrid SoK+Call for Action paper resulting from these revisions will be of significant value to the community.

**Expertise:**

Follow the literature closely, last published 5+ years ago

**Summary Of Review:**

The manuscript under consideration represents an attempt at conducting a Systematization of Knowledge (SoK) for recent (up to 2021) academic publications concerning Industrial Intrusion Detection Systems (IIDS). The work surveys a large body of works comprised of 609 publications. It outlines common trends with their evaluation methodology, particularly: the number and type of employed datasets, direct/indirect comparisons with previous results, usage of metrics, etc. The motivation behind the study emerges from the authors' concern that the sub-field of IIDS is steadily growing and, therefore, it is at risk of becoming disjoint and internally incoherent. In an attempt to highlight common issues and make a recommendation to researchers in the field under analysis, the authors conduct a Systematic Mapping Study of the IIDS's evaluation approaches used in the considered papers. At the end of the survey, the authors identify three main issues with the trends in the evaluation of IIDS and make six recommendations to enable more self-consistent and efficient research progress.

**Useful:**

yes

---

### Official Review · Reviewer_USpE · 2023-05-03

**Decision:**

Weak accept: good paper with flaws that can be fixed in three months

**Strengths:**

The paper mainly performs a qualitative analysis of existing research and thus falls into the SoK category as defined by JSys. The paper is also well written and includes effective, well-prepared figures.

The paper does a good job motivating the study by identifying a lack of empirical data about the evaluation methodology across IIDS research. The number of analyzed publications is impressive. I also liked the clear description of the publication selection process.

As someone not actively working in the area, I also appreciate the short history of the field and its development over time, the analysis of citation relationships among the selected publications, and the overview and detailed analysis of the used benchmarking datasets and evaluation metrics in Sections 4 and 5.

Finally, in Section 6, the authors succinctly summarize the findings of their study and identify recommendations for the field.

Overall, I think that readers will find the paper useful as they will learn: Which metrics are commonly used in IIDS research? What are common issues with certain metrics, and what are possible alternatives? Which benchmarking datasets are popular in the community? That releasing research artifacts as open source is necessary to facilitate meaningful performance comparisions. That one should strive to evaluate on multiple data sets. Etc. While the answers to these questions may be considered "common knowledge" the empirical data from the performed literature analysis is novel and provides concrete evidence of certain issues, while also hinting at possible ways to overcome them.


**Weaknesses:**

My two main concerns with the paper are as follows.

First, in addition to the lack of empirical data about the evaluation methodology across IIDS research, the authors motivate their work by a lack of comparability among publications. The paper essentially claims that a more coherent usage of public datasets and evaluation metrics will improve comparability and thus the state of IIDS research. On the other hand, the authors themselves note that the field of IIDS is highly heterogeneous due to the vastly different characteristics, goals, and constraints across industrial domains. Hence, in my opinion, there is a natural limit - or hinderance - to coherence and uniformity due to this inherent heterogeneity across industrial domains. I wish the authors would discuss this aspect in more detail. Perhaps, when accepting this fact, the only viable goal could be to identify recommendations for a common evaluation methodology for each individual industrial domain? To this end, the authors may consider a more fine-grained, per-domain analysis in addition to looking at the IIDS field as a whole. Another argument against coherence and uniformity, in my opinion, is that new ground-breaking research by definition needs to depart from what has been done in prior work. As such, a radically different approach may, at least to a certain extent, be incomparable to prior systems. It would be a terrible idea for a community to devalue such ground-breaking ideas by imposing the need to use a certain set of metrics and datasets in the evaluation. This is another tricky aspect that should be discussed in the beginning of the paper in order to set the right expectations.

Second, while I find the paper generally useful, the empirical data from the publication analysis novel, and the identified issues and recommendations make a lot of sense to me, I believe the ultimate findings of the study in Section 6 could have been derived, by and large, without performing the quantiative analysis of 609 publications. Many studies across various research fields have identified similar issues and recommendations, including the need for high-quality public datasets, open-source implementations, common evaluation metrics, statistically sounds ways to plan experiments and compute these metrics, and so on. In that sense, the novelty of the concrete insights and in particular the actions that the IIDS community should take moving forward is limited.

**Detailed Comments:**

A few minor comments (in no particular order) that may help further improve the paper.

Introduction: Please explain what a systematic mapping study is.

Section 2.3: What is "the KDD dataset family"?

Section 4.1.2: A citation to a prior paper does not necessarily mean that the work builds on the prior paper. While reading, I felt that the authors make this assumption here implicitly. If so, please state the assumption explicitly and clarify that you indeed checked that the work builds on the prior paper. If not, then the authors should state that their analysis reveals citations relationships, but not builds-on relationships.

Figure 3: It is hard to identify any structure or relationship from the figure. In other words, the figure felt not very useful in its current form.

I have a proposal for a better structure: It felt somewhat surprising to me that comparability and reproducibility aspects are already discussed at the end of Section 4 because an essential prerequisite for comparability and reproducibility - the evaluation metrics used - is only discussed in Section 5. It thus seems better to me 1) briefly list and describe the essential prerequisites for comparability and reproducibility, 2) then discuss and analyse each prerequisite by looking at the 609 publications (data set, metrics, availability of research artifacts, etc.), and 3) finally conclude with an overall analysis of the extent of reproducibility and comparability.

Another key prerequisite for rigorous evaluations and comparability is how the evaluation metrics are computed, which in turn relates to how the experiments are designed and executed (e.g., in order to obtain the raw data required to compute the evaluation metrics, ideally in a statistically sound way). In my opinion, this facet should also be discussed and, if possible, also extracted from the selected 609 publications.

**Expertise:**

Please contact the Area Chair if your expertise is lower than this

**Summary Of Review:**

This systemization of knowledge (SoK) paper focuses on the evaluation methodology in industrial intrusion detection systems (IIDS) research. To this end, the authors analyze 609 publications especially with regards to the used metrics and datasets as well as the extent of quantitative comparisons. The literature analysis is complemented by a taxonomy of evaluation metrics and a quantitative evaluation of the expressiveness of these metrics. The paper concludes by identifying common issues in IIDS evaluations and by giving recommendations for IIDS research to overcome these issues moving forward.


**Useful:**

yes

---

### Meta-Review · Area_Chair_jMTc · 2023-05-30

**Recommendation:** Revise
**Confidence:** 4

**Metareview:**

The main concerns for the reviewers are listed here. If the authors can address all of these issues (and the details in the full reviews) in the given timeframe, then the paper has a high chance at acceptance.

**Pros**

- well written
- comprehensive analysis of lots of papers (600+ papers!)
- Presents a convincing, comprehensive taxonomy

**Cons**

- the recommendations in section 6 are obvious and don’t provide too many insights
- the paper has more of a survey feel than an SoK — the objective of the latter is to educate both, experts and new readers but a reader would still be unaware of the major techniques used in IIDS (or even some of the prominent works in this domain) after reading this
- the heterogeneity of different industries is not captured well — the authors need to set the right expectations for metrics, comparison across industries, can common solutions be found
- the novelty of the concrete insights and in particular the actions that the IIDS community should take moving forward is limited
- a citation to a prior paper does not necessarily mean that the work builds on the prior paper; the authors make this assumption here implicitly. If so, please state the assumption explicitly and clarify that you indeed checked that the work builds on the prior paper. If not, then the authors should state that their analysis reveals citations relationships, but not builds-on relationships
- figure 3, in its current form, doesn’t help identify any structure or useful relationships
- the paper focuses purely on the evaluation metrics without any ties to the technique used

**Additional suggestions on how to improve the paper:**

- the section on Reproducibility and Comparability would be useful if it was discussed after the survey on evaluation metrics.
- to align it better towards an SoK, the authors could complement the manuscript with additional sections that provide an actual assessment of the recent advancements in the field of IIDS by building on top of what the cited surveys [49] and [75] already do.
- identify future research directions or challenges that require the community attention
- it is still important to explain at least one case study on IIDS and the resulting dataset in detail to the reader. Currently, the paper does not contain any of it. It assumes that the reader is familiar with at least the main ideas in IIDS research, which may not be the case.
- the paper can instead focus on a couple of case studies and explain the resulting datasets in detail (this could be Morris-Gas and SWaT), and then spend the majority of sections on evaluating and discussing different IIDSs and metrics on these and qualitatively explaining the results. Hence, I believe Sections 5.3 and 6 should be the main focus of SoK. Everything else seems like a survey paper.
- the authors may consider a more fine-grained, per-domain analysis in addition to looking at the IIDS field as a whole.

---

### Decision · Program_Chairs · 2023-05-22

**Decision:**

Revise

**Comment:**

Dear author,

Thank you for submitting your work to JSys. The reviewers agree that the work is valuable and should get published, but also formulated a number of concerns and recommendations to help make the manuscript stronger. Please read through the meta-review carefully.

Therefore, the area chair recommends a "Revise" decision. Information on the revisions is available on the JSys website: https://www.jsys.org/instructions#submitting-a-revision
Please do not hesitate to contact the Editors-in-Chief if anything is unclear.

We look forward to receiving your revised manuscript.

[Please accept our sincere apologies for the delays in the procedure]